# eIF3 engages with 3'-UTR termini of highly translated mRNAs

Santi Mestre-Fos[1,2], Lucas Ferguson[2,3], Marena I Trinidad[1,4], Nicholas T Ingolia[2,5], Jamie HD Cate[1,2,6]*

[1]Innovative Genomics Institute, University of California, Berkeley, Berkeley, United States; [2]Department of Molecular and Cell Biology, University of California, Berkeley, Berkeley, United States; [3]Center for Computational Biology, University of California, Berkeley, Berkeley, United States; [4]Howard Hughes Medical Institute, University of California, Berkeley, Berkeley, United States; [5]California Institute for Quantitative Biosciences, University of California, Berkeley, Berkeley, United States; [6]Department of Chemistry, University of California, Berkeley, Berkeley, United States

## eLife Assessment

This **valuable** study shows previously unappreciated binding of the eukaryotic translation initiation factor 3 (eIF3) to the poly(A) tail proximal portion of 3' untranslated regions (UTRs) of mRNAs that are efficiently translated in neuronal progenitors. The authors' conclusions are supported by **solid** experimental evidence which is based on several orthogonal systems biology approaches. This article is of considerable interest to the broad spectrum of biomedical researchers interested in studying post-transcriptional regulation of gene expression.

**\*For correspondence:**
j-h-doudna-cate@berkeley.edu

**Competing interest:** The authors declare that no competing interests exist.

## Abstract

Stem cell differentiation involves a global increase in protein synthesis to meet the demands of specialized cell types. However, the molecular mechanisms underlying this translational burst and the involvement of initiation factors remains largely unknown. Here, we investigate the role of eukaryotic initiation factor 3 (eIF3) in early differentiation of human pluripotent stem cell (hPSC)-derived neural progenitor cells (NPCs). Using Quick-irCLIP and alternative polyadenylation (APA) Seq, we show eIF3 crosslinks predominantly with 3' untranslated region (3'-UTR) termini of multiple mRNA isoforms, adjacent to the poly(A) tail. Furthermore, we find that eIF3 engagement at 3'-UTR ends is dependent on polyadenylation. High eIF3 crosslinking at 3'-UTR termini of mRNAs correlates with high translational activity, as determined by ribosome profiling, but not with translational efficiency. The results presented here show that eIF3 engages with 3'-UTR termini of highly translated mRNAs, likely reflecting a general rather than specific regulatory function of eIF3, and supporting a role of mRNA circularization in the mechanisms governing mRNA translation.

## Introduction

Stem cells are a group of diverse cells that are characterized by their ability to self-renew and differentiate into multiple cell types. This remarkable plasticity enables them to contribute to tissue development, maintenance, and repair. In their quiescent state, stem cells present low protein synthesis levels, conserving energy and resources while maintaining their undifferentiated state. However, upon receiving differentiation signals, they exhibit a global increase in protein synthesis that results in a drastic change in the proteome composition to meet the demands of newly specialized progenitor cells (*Baser et al., 2019*; *Blanco et al., 2016*; *Saba et al., 2021*; *Sampath et al., 2008*; *Signer et al., 2014*; *Zismanov et al., 2016*). Since initiation is translation's rate-limiting step, it must be highly

regulated to maintain quiescence and to promote the global increase in translation that occurs during the initial steps of stem cell differentiation. Despite its importance, the roles that initiation factors play in quiescent and newly differentiated stem cells are poorly understood.

eIF3 is the largest of all initiation factors in eukaryotes and plays a pivotal role in the initiation of translation. In humans, eIF3 is composed of 13 subunits and helps position ribosomes at the start codon of mRNAs (*Jackson et al., 2010*; *Figure 1A*). However, recent discoveries have highlighted several non-canonical functions of eIF3 that extend beyond its general role in translation initiation. For instance, photoactivatable ribonucleoside-enhanced crosslinking and immunoprecipitation (PAR-CLIP) experiments (*Hafner et al., 2010*) performed in human embryonic kidney (HEK) 293T cells uncovered regulatory roles of eIF3 in the translation of a specific pool of mRNAs (*Lee et al., 2015*). These studies also revealed that eIF3 subunit d (EIF3D) activates the translation of *JUN* mRNA by binding to its 5' cap (*Lee et al., 2016*). Intriguingly, the pool of mRNAs regulated by eIF3 varies dramatically across different cell types and physiological conditions. PAR-CLIP experiments performed in Jurkat T cells showed that robust T cell activation requires the direct interaction of eIF3 with the mRNAs encoding for the human T cell receptor subunits α and β, a phenomenon that was also observed in primary T cells (*De Silva et al., 2021*). Additionally, eIF3 modulates the translation of mRNAs critical for adaptation and survival under stress conditions such as nutrient deprivation (*Lamper et al., 2020*). Beyond its established functions in translation initiation, eIF3 has been shown to also be involved in translation elongation (*Lin et al., 2020*; *Wagner et al., 2020*) and termination (*Beznosková et al., 2013*; *Valásek, 2012*; *Valášek et al., 2017*). These findings indicate that eIF3 serves distinct regulatory roles depending on specific cellular environments.

Here, we derived NPCs from hPSCs and committed them to a forebrain neuron fate to study the potential roles of eIF3 in regulating the early differentiation-dependent global increase in protein synthesis. NPCs are considered more committed to the neural lineage than NSCs (*Oikari et al., 2016*) and hence are likely to present higher protein synthesis levels than quiescent NSCs (*Baser et al., 2019*). Despite their commitment, NPCs maintain the self-renewal and multipotent characteristics of NSCs and, therefore, are widely used to study stem-like properties in vitro. Our study focused on identifying the mRNAs that crosslink to eIF3 during the translational burst observed upon NPC differentiation. Instead of identifying specific regulatory roles for eIF3, we discovered that eIF3 predominantly crosslinks to the 3'-UTRs of mRNAs. Using next-generation sequencing to map polyadenylation events in NPCs, we found that eIF3 crosslinking at 3'-UTRs occurs at the termini of multiple mRNA isoforms, adjacent to polyadenylated (polyA) tails. Transcriptome-wide analysis further demonstrated that increased eIF3 crosslinking at 3'-UTR termini correlates with an increased level of ribosome footprint (RPF) levels on mRNAs. Interestingly, the binding of eIF3 to 3'-UTR termini was observed only in polyadenylated transcripts and is independent of interactions between eIF3 and polyA-binding proteins, as inferred from eIF3 immunoprecipitation experiments. These findings show that high levels of active translation drive eIF3 engagement at mRNA 3'-UTR termini. Given eIF3's known role in associating with 5'-UTRs to promote translation initiation, our observation of eIF3 interactions with 3'-UTR ends points to an unexpected role for eIF3 in mRNA circularization, potentially facilitating communication between the 5' and 3' ends of actively translated mRNAs.

## Results

### Generation of hPSC-derived NPCs

We differentiated hPSCs toward NPCs by embryoid body (EB) formation followed by neural rosette selection, giving rise to NPCs after ~3 wk (*Figure 1B*). Successful NPC differentiation was assessed by western blotting. NPCs cultured for several passages (up to passage 7) presented an increase in the expression of neural markers Pax6 and Sox1 and the loss of the pluripotent marker Oct4 (*Figure 1C*).

### Early NPC differentiation results in a global increase in protein synthesis

To find the translational burst that occurs during the initial steps of stem cell differentiation, we differentiated NPCs toward a forebrain neuron fate using a dual-SMAD inhibition protocol (*Chambers et al., 2009*) and collected them at different time points that ranged from 1 hr to several days. In order to study active translation levels, prior to cell collection, we treated cells for a short time

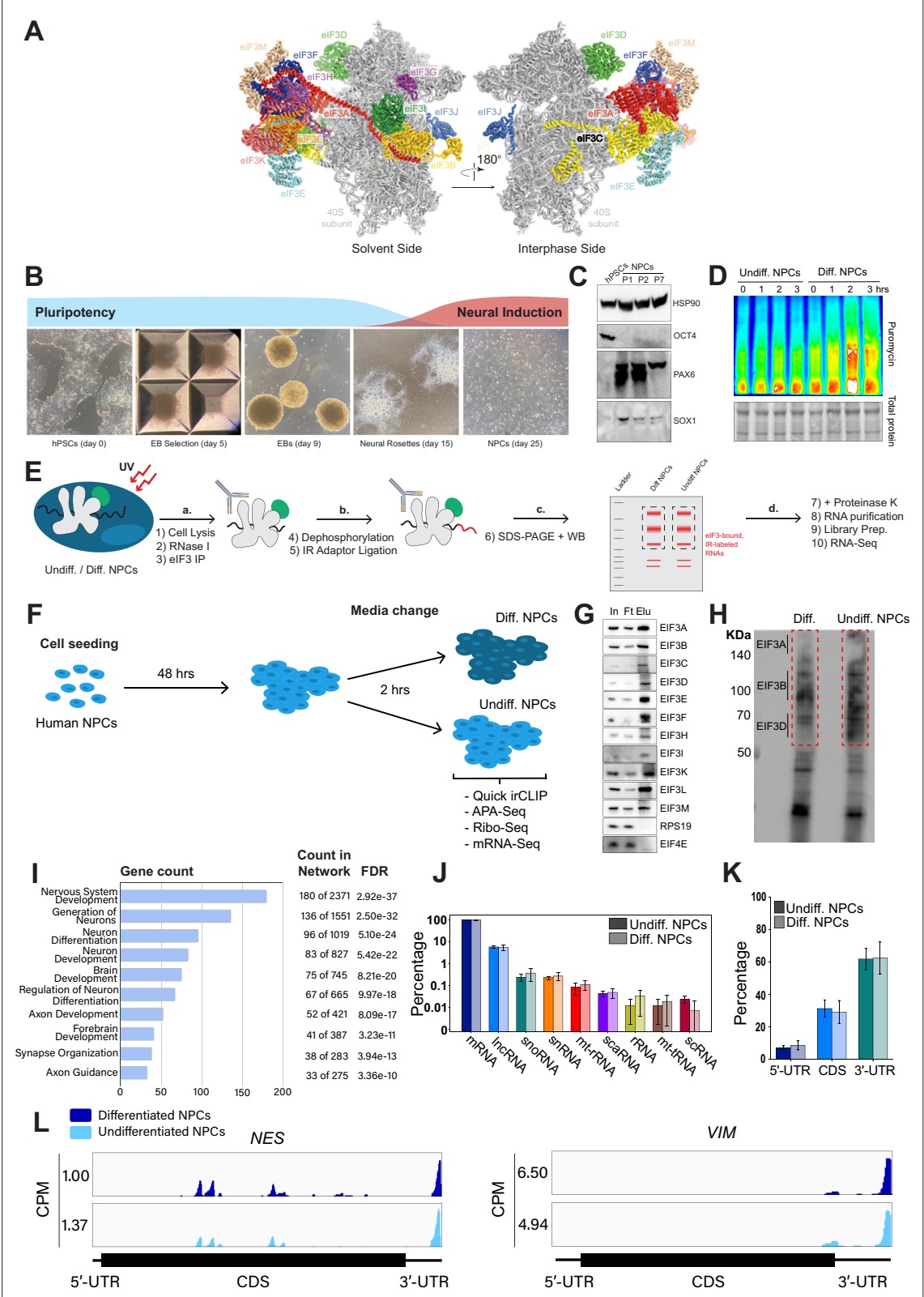

**Figure 1.** Analysis of Quick irCLIP of eukaryotic initiation factor 3 (eIF3) to RNAs in undifferentiated and differentiated neural progenitor cells. (**A**) Structure of the human eIF3 complex as part of the 48 S initiation complex. PDB: 6ZMW (*Brito Querido et al., 2020*). (**B**) Generation of neural progenitor cells (NPCs) from human pluripotent stem cells (hPSCs) by embryoid body (EB) and neural rosette selections. (**C**) Western blots of neural markers Pax6 and Sox1 and pluripotent marker Oct4 from hPSCs and hPSC-derived NPCs (NPCs corresponding to passages 1, 2, and 7 are shown).

*Figure 1 continued on next page*

*Figure 1 continued*

(**D**) Western blots of puromycin-treated (15 min) NPCs. Total protein stain is shown as loading control. Differentiated NPCs were treated with forebrain neuron differentiation medium for the indicated time points. Undifferentiated NPCs were treated with NPC medium for the indicated time points. The ratios of puromycin signal to total protein levels can be found in *Figure 1—figure supplement 1A*. (**E**) Schematic of Quick-irCLIP. (**F**) Schematic of the NPC treatment performed in the Quick-irCLIP, polyadenylation sequencing (APA-seq), Ribosome profiling, and mRNA-Seq experiments. Treatment consisted of change of media and incubation at 37 °C for 2 hrs. Differentiation media was used for differentiated (Diff.) NPCs and Basal media for undifferentiated (Undiff.) NPCs. (**G**) Immunoprecipitation samples of assembled eIF3 complexes from undifferentiated NPCs. 5% inputs (In), flowthroughs (Ft), and eluates (Elu) are shown. Western blot for subunit EIF3G is not shown because we could not identify an effective antibody for its detection. EIF3J, not shown, is usually dissociated from the eIF3 complex upon immunoprecipitation. However, we detected EIF3G by mass spectrometry in the IPs (*Supplementary file 1*). (**H**) Infrared (IR) image of IR dye-labeled, eIF3 UV-crosslinked RNA transcripts from Diff. and Undiff. NPCs. Regions marked with red boxes, which correspond to subunits EIF3A through EIF3D, were excised from the blot. (**I**) Biological Function enrichment determined using the STRING database for the undifferentiated NPC biological replicates of the EIF3A/B/C/D Quick-irCLIP libraries, using the top 500 mRNA hits from the crosslinking analysis. (**J**) Categories of RNAs crosslinked to eIF3 in undifferentiated and differentiated NPCs, for three replicates plotted in log-scale with standard deviation error bars. (**K**) mRNA regions that crosslink to eIF3 in undifferentiated and differentiated NPCs. (**L**) Crosslinking of eIF3 across the transcripts of *NES* and *VIM* mRNAs in differentiated and undifferentiated NPCs. Read coverage is provided in Counts Per Million (CPM).

The online version of this article includes the following source data and figure supplement(s) for figure 1:

**Source data 1.** Files of original western blots for panels C and G, along with gels of the eukaryotic initiation factor 3 (eIF3) crosslinked RNA in the Quick-irCLIP experiment in panel H.

**Source data 2.** Files of original blots and gels in panels C, G, and H, with experimental conditions marked.

**Figure supplement 1.** Neural progenitor cells (NPCs) treated with differentiation media or kept in undifferentiated media (2 hr treatment).

**Figure supplement 1—source data 1.** Original gel with total protein staining and anti-puromycin western blot in panel B.

**Figure supplement 1—source data 2.** Original gel with total protein staining and anti-puromycin western blot in panel B with experimental conditions marked.

**Figure supplement 2.** Venn Diagram of genes identified by Quick-irCLIP (differentiated neural progenitor cells, NPCs) and PAR-CLIP (HEK293T and activated Jurkat T cells).

**Figure supplement 3.** Crosslinking of eukaryotic initiation factor 3 (eIF3) in undifferentiated neural progenitor cells (NPCs) and in HEK293T cells.

**Figure supplement 4.** Crosslinking of eukaryotic initiation factor 3 (eIF3) in differentiated and undifferentiated NPCs across the 3'-UTR regions of *ACTG1* and *FTL* mRNAs.

**Figure supplement 5.** Titration of RNAse I for the Quick-irCLIP experiment.

**Figure supplement 5—source data 1.** Original gels of labeled RNAs from the eukaryotic initiation factor 3 (eIF3) pulldowns.

**Figure supplement 5—source data 2.** Original gels of labeled RNAs from the eukaryotic initiation factor 3 (eIF3) pulldowns, with conditions indicated.

**Figure supplement 6.** Correlation of Quick-irCLIP replicates.

with puromycin, a tRNA analog that gets incorporated into the C-terminus of elongating nascent chains, releasing them from translating ribosomes. By immunoblotting with puromycin antibodies, puromycin-containing nascent peptides can be detected, reporting on active levels of protein translation. Our western blotting results show that NPCs treated for 2 hr with forebrain neuron differentiation medium present a significant increase in puromycin incorporation into nascent chains compared to NPCs treated for the same amount of time with medium that keeps them in their undifferentiated state (*Figure 1D*, *Figure 1—figure supplement 1A,B*), indicating a differentiation-dependent global increase in protein synthesis.

## eIF3 crosslinks to 5'- and 3'-UTRs of neurologically-relevant mRNAs

To identify the RNA transcripts that interact with eIF3 during the observed translational burst in differentiated NPCs, we performed Quick-irCLIP (*Kaczynski et al., 2019*; *Figure 1E*). We selected the aforementioned 2 hr treatment timepoint and immunoprecipitated eIF3 complexes from early differentiated NPCs that had started the differentiation pathway toward forebrain neurons (differentiated NPCs) and NPCs that we kept in their progenitor state for the same amount of time (undifferentiated NPCs) (*Figure 1F*). Differentiation treatment does not result in major cellular morphology changes (*Figure 1—figure supplement 1C*). After UV crosslinking the cells, we treated cell lysates with RNAse I, followed by eIF3 immunoprecipitation, dephosphorylation of the protein-bound transcripts, IR adaptor ligation, and RNA-protein complex visualization via SDS-PAGE. We assessed the success of eIF3 immunoprecipitation by western blotting (*Figure 1G*) and mass spectrometry (*Supplementary file 1*). The previous PAR-CLIP studies performed with the eIF3 complex in HEK293T and Jurkat cells

identified eIF3 subunits EIF3A, EIF3B, EIF3D and, to a lesser extent, EIF3G as the four subunits of the complex presenting significant amounts of RNA crosslinks (*De Silva et al., 2021*; *Lee et al., 2015*). Consequently, here we excised the RNA smears that appeared in the regions of subunits EIF3A/B/C/D (from ~170 kDa to ~65 KDa) (*Figure 1H*).

Using the STRING database (*Szklarczyk et al., 2019*) for the top 500 transcripts that crosslink to eIF3, we observed highly similar sets of mRNAs in undifferentiated and differentiated NPCs (449 in common). We observe a significant enrichment in neurologically-relevant biological processes, such as 'generation of neurons,' 'neuron differentiation,' 'neuron projection development,' and 'axon development' (*Figure 1I*). To determine which top eIF3-binding transcripts are common between NPCs and the previous Jurkat T cell (*De Silva et al., 2021*) and HEK293T cell (*Lee et al., 2015*) PAR-CLIP studies, we compared the top ~200 eIF3-crosslinked transcripts between the three cell lines. We found that only 12 are common between NPCs and Jurkat T cells and only eight are common between NPCs and HEK293T, with no common hits between the three cell types (*Figure 1—figure supplement 2*).

Both in differentiated and undifferentiated NPCs, eIF3 predominantly (>90%) crosslinks to mRNAs over other non-coding RNAs such as lncRNAs and snoRNAs (*Figure 1J*). Within mRNA regions, eIF3 primarily interacts with 3'-UTRs and, to a lesser extent, with CDS regions and 5'-UTRs (*Figure 1K*). Our Quick-irCLIP experiment in NPCs identified several eIF3 crosslinks located at 5'-UTR regions that were also observed in the previous eIF3 PAR-CLIP experiment performed in HEK293T (*Lee et al., 2015*), such as *CCND2* and *TUBB* mRNAs (*Figure 1—figure supplement 3*). The distinctive 'pan-mRNA' pattern observed in the eIF3 PAR-CLIP performed in Jurkat T cells (*De Silva et al., 2021*) is not observed in NPCs. Notably, here we observe an enrichment of eIF3 crosslinks at 3'-UTRs of transcripts encoding neurologically-relevant proteins, such as *VIM* and *NES* (*Figure 1L*), as well as highly abundant mRNAs, such as *ACTG1* and *FTL* (*Figure 1—figure supplement 4*). Overall, we observe similar eIF3 crosslinking levels to transcripts in differentiated and undifferentiated NPCs (*VIM* and *NES* are shown as examples in *Figure 1L*).

## eIF3 3'-UTR crosslinking events map to 3'-UTR termini of multiple isoforms, upstream of poly(A) tails

In many instances, we observe eIF3 crosslinking peaks at the termini of annotated 3'-UTRs, adjacent to the poly(A) tail and where the polyadenylation signal (PAS; canonical PAS sequence = AAUAAA) is located (*Figure 2A*, *TUBB*). However, in addition to ends of annotated 3'-UTRs, we also observe the same type of crosslinking pattern across multiple regions within 3'-UTRs (*Figure 1L*, *VIM*). Looking at the nucleotide sequence of the regions within 3'-UTRs where eIF3 crosslinks, we observe that most of them contain PAS sequences as well (*Figure 2A and B*, *TUBB* and *APP*). Hypergeometric optimization of motif enrichment (HOMER) motif discovery analysis (*Heinz et al., 2010*) of the eIF3 crosslinks that map to 3'-UTR regions revealed that in differentiated and undifferentiated NPCs 58% and 55% of the 3'-UTR eIF3 crosslinks, respectively, map to regions that are extremely enriched in the canonical PAS sequence AAUAAA (*Figure 2C*, *Supplementary file 2*, *Supplementary file 3*). We also find other enriched 3'-UTR motifs but none of them has either the level of eIF3 enrichment nor the statistical significance as AAUAAA (*Figure 2—figure supplement 1*). Overall, these data indicate eIF3 crosslinks near 3'-UTR termini of multiple isoforms, upstream of poly(A) tails in NPCs. In neurons, 3'-UTR isoforms are known to play key roles in mRNA localization and expression in different neuronal compartments (*Taliaferro et al., 2016*; *Tushev et al., 2018*) and alternative polyadenylation is highly prevalent in the nervous system, leading to the expression of multiple mRNA isoforms with different 3'-UTR lengths (*Miura et al., 2013*).

To test our hypothesis that the observed eIF3 crosslinking pattern reflects the expression of bona fide alternative 3'-UTRs in NPCs, we performed alternative polyadenylation next-generation sequencing (APA-Seq) in undifferentiated NPCs. Given that we observe similar eIF3 crosslinking levels in undifferentiated and differentiated NPCs, we did not perform APA-Seq in differentiated NPCs. APA-Seq creates sequencing libraries by amplifying the region upstream of a polyadenylation event, allowing the identification of all 3'-UTR mRNA isoforms being expressed for any given gene. Comparing our Quick-irCLIP and APA-Seq experiments, we observe that the eIF3 peaks located in 3'-UTRs map to the same location as APA peaks, indicating that the predominant eIF3 crosslinking peaks in 3'-UTRs identified by Quick-irCLIP localize to 3'-UTR termini, adjacent to the poly(A) tail. For instance, the eIF3 crosslinking peaks located in the 3'-UTR of the *MAP1B* mRNA correspond to APA-Seq peaks

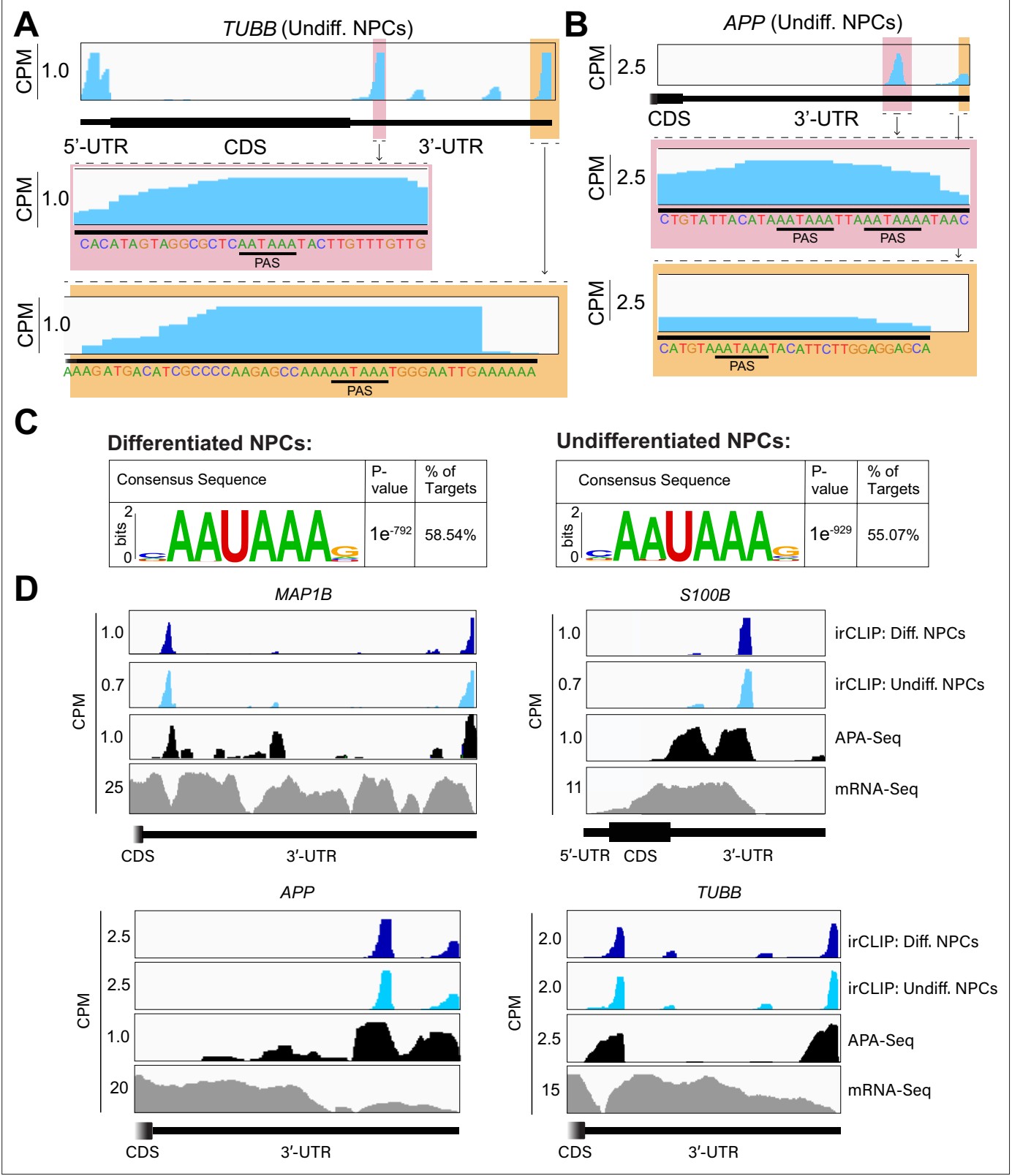

**Figure 2.** Crosslinking of eukaryotic initiation factor 3 (eIF3) to mRNA 3'-UTRs adjacent to the poly(A) tail. (**A**) Crosslinking of eIF3 across the *TUBB* mRNA in undifferentiated neural progenitor cells (NPCs). A zoomed-in view of *TUBB* mRNA 3'-UTR terminus with the eIF3 crosslinks is shown. The polyadenylation signal (PAS) is marked. (**B**) Crosslinking of eIF3 across the *APP* mRNA 3'-UTR in undifferentiated NPCs. A zoomed-in view of the two 3'-UTR regions presenting eIF3 crosslinks are shown as well as their PAS sequences. (**C**) Sequence logo of the eIF3 crosslinks located at 3'-UTRs in

*Figure 2 continued on next page*

*Figure 2 continued*

differentiated and undifferentiated NPCs. (**D**) Crosslinking of eIF3, polyadenylation sequencing (APA-seq) peaks, and mRNA-seq across the *MAP1B, S100B, TUBB,* and *APP* mRNAs in differentiated and undifferentiated NPCs.

The online version of this article includes the following figure supplement(s) for figure 2:

**Figure supplement 1.** Motif enrichment of eukaryotic initiation factor 3 (eIF3) 3'-UTR crosslinks for undifferentiated and differentiated neural progenitor cells (NPCs).

**Figure supplement 2.** Examples of eukaryotic initiation factor 3 (eIF3) crosslinking to 3'-UTR regions of mRNAs.

(*Figure 2D*). The aforementioned cases of *NES* and *VIM* mRNAs, where we observed eIF3 crosslinks in their 3'-UTRs, also have APA-Seq peaks at the same locations as the irCLIP peaks (*Figure 2—figure supplement 2*). As another example, our irCLIP experiment shows that eIF3 crosslinks to both the 5'- and 3'-UTRs of the *S100B* mRNA, which encodes for a calcium-binding protein highly expressed in brain cells and that is commonly used as a glial marker (*Brockes et al., 1979*; *Ferri et al., 1982*; *Ludwin et al., 1976*). Within *S100B*'s 3'-UTR, we observe 2 major eIF3 crosslinking peaks at the 3'-UTR termini of 2 distinct *S100B* mRNA isoforms expressed in NPCs (*Figure 2D*). Interestingly, we found that eIF3 exhibits different crosslinking levels to the 3'-UTR termini of various 3'-UTR isoforms that are similarly expressed. For instance, our APA-Seq data reveals two major *S100B* mRNAs with similar expression levels. However, eIF3 preferentially crosslinks to the distal 3'-UTR isoform rather than the proximal 3'-UTR isoform (*Figure 2D*). Conversely, in the case of *APP*, which also expresses 2 major mRNA isoforms, eIF3 predominantly binds to the 3'-UTR terminus of the proximal isoform. Finally, similar to the case of *MAP1B* mRNA, in *TUBB* mRNA the extent of eIF3 crosslinking correlates with mRNA isoform expression, indicating that eIF3 binds to both major *TUBB* mRNA isoforms to a similar extent (*Figure 2D*). Overall, these results show that the eIF3 crosslinking events that we observe in 3'-UTRs often occur at 3'-UTR termini of multiple mRNA isoforms, upstream of poly(A) tails.

## eIF3 engages with 3'-UTR termini upon increased levels of protein synthesis

Our Quick-irCLIP experiment showed that eIF3 interacts with the 3'-UTR termini of mRNAs expressed in both differentiated and undifferentiated NPCs. Interestingly, eIF3 interacts with the mRNA encoding the transcription regulator ID2 only upon NPC differentiation (*Figure 3A*). ID2 is involved in migrating NPC differentiation into olfactory dopaminergic neurons (*Havrda et al., 2008*). Therefore, it is possible that the observed eIF3 engagement at its 3'-UTR terminus in differentiated NPCs is due to *ID2*'s mRNA translation activation upon differentiation. This would suggest that eIF3 interactions with 3'-UTR termini are involved with active translation levels for a given transcript.

To test whether eIF3 engagement at 3'-UTR termini correlates with translation activity at a transcriptome-wide level, we performed ribosome profiling, a technique that is based on deep sequencing of ribosome-protected mRNA fragments (RPFs) (*Ingolia et al., 2011*), in undifferentiated and differentiated NPCs. Here, we used a low-bias ribosome profiling approach that takes advantage of the enzymatic activities of reverse transcriptase (RT) from eukaryotic retroelements to reduce the technicalities associated with ribosome profiling (*Ferguson et al., 2023*). In addition, this method uses P1 nuclease instead of the commonly used RNase I and replaces RNA ligation with Ordered Two-Template Relay (OTTR) (*Upton et al., 2021*). P1 nuclease digestion not only preserves monosome-protected mRNA fragments, which are the commonly used fragments in ribosome profiling experiments, but also mRNA fragments protected by collided ribosomes (disomes), which report on ribosome stalling and provide a screenshot of ribosome quality control (RQC) pathways (*Ferguson et al., 2023*; *Meydan and Guydosh, 2020*).

Here, we deep sequenced monosome- and disome-protected fragments from undifferentiated and differentiated NPCs (*Figure 3—figure supplement 1*, *Figure 3—figure supplement 2*). We observe that about a third (monosome) and about a fifth (disome) of the libraries mapped to mRNAs (*Figure 3—figure supplement 2A*), similar to previous reports (*Ferguson et al., 2023*). P1 nuclease digestion results in monosome footprints of 32–40 nts in length, observed in both differentiated and undifferentiated NPCs (*Figure 3—figure supplement 2B*). We also observe two populations of footprints in the disome library as previously reported upon P1 digestion, with the larger one corresponding to the 'true disome' population (i.e. ≥60 nt disome footprints shown in *Figure 3—figure*

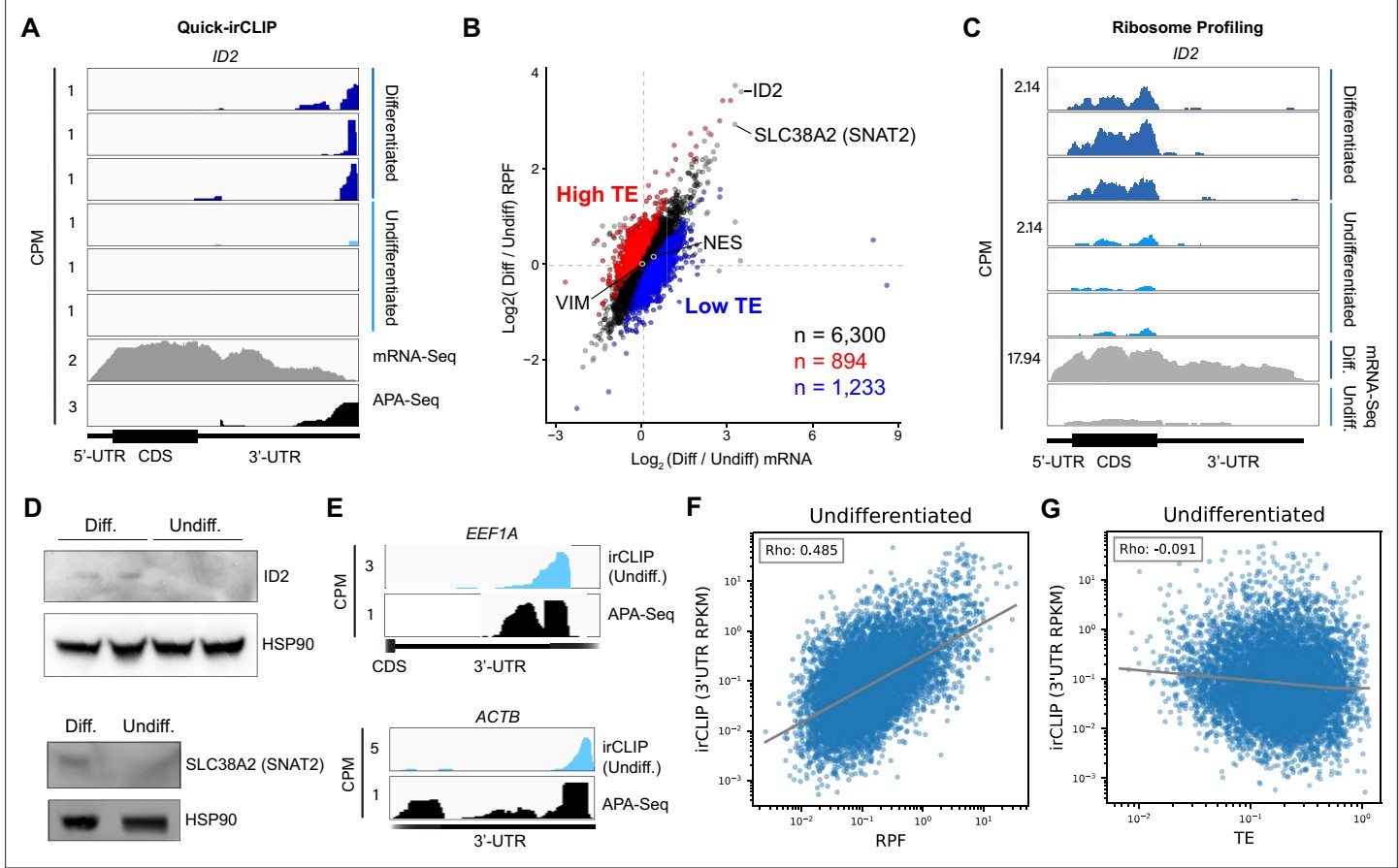

**Figure 3.** Ribosome profiling of undifferentiated and differentiated neural progenitor cells (NPCs), and comparisons to eukaryotic initiation factor 3 (eIF3) Quick irCLIP crosslinking. (**A**) Crosslinking of eIF3 across the *ID2* mRNA in undifferentiated and differentiated NPCs. Polyadenylation sequencing (APA-seq) and mRNA-seq data are also shown. (**B**) $\log_2$ fold change of ribosomal footprints and $\log_2$ fold change of mRNA transcript levels upon NPC differentiation. Transcripts with a higher TE in differentiated NPCs compared to undifferentiated NPCs (high TE) are indicated in blue. Transcripts with a lower TE in differentiated NPCs compared to undifferentiated NPCs (low TE) are shown in red. The number of samples within each condition is shown. (**C**) Ribosomal footprints across the *ID2* mRNA in undifferentiated and differentiated NPCs. mRNA-seq data are also shown. (**D**) Western blots of ID2, SLC38A2 (SNAT2), and Hsp90 in undifferentiated and differentiated NPCs. (**E**) Crosslinking of eIF3 to the 3'-UTR of the *EEF1A* and *ACTB* mRNAs in undifferentiated NPCs. APA-seq data is also shown. (**F**) Comparison of irCLIP reads in mRNA 3'-UTRs to ribosome protected fragments (RPF) in undifferentiated NPCs. (**G**) Comparison of irCLIP reads in mRNA 3'-UTRs to translation efficiency (TE) in undifferentiated NPCs. Outliers exceeding the TE's 99th percentile were removed, to make the log-log plot more readable. In panels (**F**) and (**G**), the Spearman correlation of average irCLIP 3'-UTR RPKM with mean RPF or TE across replicates (n=3) is shown.

The online version of this article includes the following source data and figure supplement(s) for figure 3:

**Source data 1.** Original western blots for *Figure 3D*.

**Source data 2.** Original western blots for *Figure 3D* with neural progenitor cell (NPC) treatment conditions indicated.

**Figure supplement 1.** TBE 8% Urea-PAGE of the Ordered Two-Template Relay (OTTR) reaction products of cDNA libraries constructed from undifferentiated and differentiated neural progenitor cells (NPCs).

**Figure supplement 1—source data 1.** Original gels of Ordered Two-Template Relay (OTTR) reaction products used for ribosome profiling cDNA library construction.

**Figure supplement 1—source data 2.** Original gels of Ordered Two-Template Relay (OTTR) reaction products used for ribosome profiling cDNA library construction, with locations of regions cut out of gel indicated.

**Figure supplement 2.** Ribosome profiling of undifferentiated and differentiated neural progenitor cells (NPCs).

**Figure supplement 3.** SLC38A2 expression in differentiated and undifferentiated neural progenitor cells (NPCs).

**Figure supplement 3—source data 1.** Original western blot for *Figure 3—figure supplement 3A*.

**Figure supplement 3—source data 2.** Original western blot for *Figure 3—figure supplement 3A* with conditions indicated.

**Figure supplement 4.** Correlation of Quick-irCLIP with Ribosome Profiling and Translational Efficiency in Differentiated neural progenitor cells (NPCs).

supplement 2C), composed of colliding ribosomes and populations of two translating ribosomes in close proximity to each other, and to the 'sub-disome' population (i.e. <60 nt disome footprints shown in *Figure 3—figure supplement 2C*), proposed to correspond to 80 S monosomes in close proximity to a scanning 40 S ribosomal subunit during translation initiation or to a 40 S ribosomal subunit from which a 60 S ribosomal subunit was recently dissociated from during translation termination (*Ferguson et al., 2023*). In agreement with the translational burst that we identified (*Figure 1D*), our monosome profiles show an increase in RPFs around the start codon in differentiated NPCs, indicating an increase of initiating ribosomes in differentiated NPCs with respect to the undifferentiated conditions (*Figure 3—figure supplement 2D*). We observed a global increase in ribosome traffic after differentiation as evidenced by the increase in true-disome footprints among CDS and terminating codons (*Figure 3—figure supplement 2C*). By DESeq2 analysis, which does not consider transcript length in read-count normalization, we found in both differentiated and undifferentiated NPCs an increase in true-disome CDS occupancy, relative to monosome, correlates with CDS length (*Figure 3—figure supplement 2E*). This was expected since longer coding sequences are more likely to have more translating ribosomes per transcript than shorter ones.

Our ribosome profiling experiment indicates that for most mRNAs (n=6300 of a total of 8427, i.e. ~75%), translation efficiency levels (RPF counts/mRNA counts) remain unchanged upon NPC differentiation (*Figure 3B*, **Supplementary file 4**,). *ID2* is one of the transcripts with the highest increase in both RPFs and mRNA counts upon NPC differentiation (*Figure 3B*). Specifically, we observe a >fivefold RPF count increase upon NPC differentiation (*Figure 3C*), indicating a significant burst in *ID2* mRNA translation activity. However, we also observe that NPC differentiation results in a significant increase in *ID2* mRNA transcript levels (~15 fold, *Figure 3C*). Western blots of ID2 and a second protein SLC38A2 (SNAT2) with increased RPFs on its mRNA in differentiated vs. undifferentiated NPCs are in agreement with the ribosome profiling data (*Figure 3D*, *Figure 3—figure supplement 3*). Additional western blots for SLC38A2 can be found in *Figure 3—figure supplement 3*. Notably, this observed stepwise increase in *ID2* mRNA levels and translation is well corroborated by our observed increase in eIF3 crosslinking to *ID2* mRNA's 3'-UTR termini, suggesting a connection for eIF3 in maintaining and promoting *ID2* translation upon differentiation beyond the 5'-UTR.

We next looked at the most actively translated transcripts in NPCs by sorting the ribosome profiling hits based on their amount of RPFs. We noticed that the most actively translated transcripts, such as *EEF1A* and *ACTB*, also present significant eIF3 peaks at their 3'-UTR termini (*Figure 3E*). Together with the *ID2* mRNA results, these data suggest that eIF3 interacts with 3'-UTR ends of mRNAs being actively translated.

To test whether the observed increase in eIF3 crosslinking to 3'-UTRs is linked to an increase in ribosome translation at a global level, we performed DESeq2 on the counted 3'-UTR eIF3 irCLIP reads to genes with 3'-UTRs longer than 50 nts. Indeed, regardless of whether we examined undifferentiated or differentiated NPCs, we found genes which had an increase in 3'-UTR irCLIP crosslinking tended to have an increase in ribosome occupancy (*Figure 3F*, *Figure 3—figure supplement 4A*). By contrast, the increase in 3'-UTR irCLIP crosslinking correlated less well with increases in translational efficiency (*Figure 3G*, *Figure 3—figure supplement 4B*). Overall, our ribosome profiling data combined with our eIF3 irCLIP data suggest that, in both undifferentiated and differentiated NPCs, there is a global increase in eIF3 engagement at 3'-UTR termini that correlates with an overall increase in protein synthesis.

## eIF3 interactions with 3'-UTR termini requires polyadenylation but is independent of poly(A) binding proteins

eIF3 crosslinking at 3'-UTR termini suggests that it must be in close proximity to the poly(A) tail and poly(A) tail binding proteins (PABPs). To determine whether the presence of eIF3 at 3'-UTR ends requires polyadenylation, we looked at histone mRNAs. Canonical histone mRNAs are the only known cellular non-polyadenylated mRNAs in eukaryotes due to their cell-cycle dependency and their need to be degraded rapidly. On the other hand, several variant histone mRNAs are not cell-cycle dependent and are polyadenylated. In non-polyadenylated canonical histone mRNAs (*H2AC11*, *H3C1*, and *H4-16*), we see no traces of eIF3 crosslinking at 3'-UTR termini nor APA events, as expected (*Figure 4A*, *Figure 4—figure supplement 1A*). However, for the variant polyadenylated histone mRNAs (*H3-3B* and *H2AZ1*), we observe eIF3 crosslinks that localize with APA-Seq peaks (*Figure 4A*,

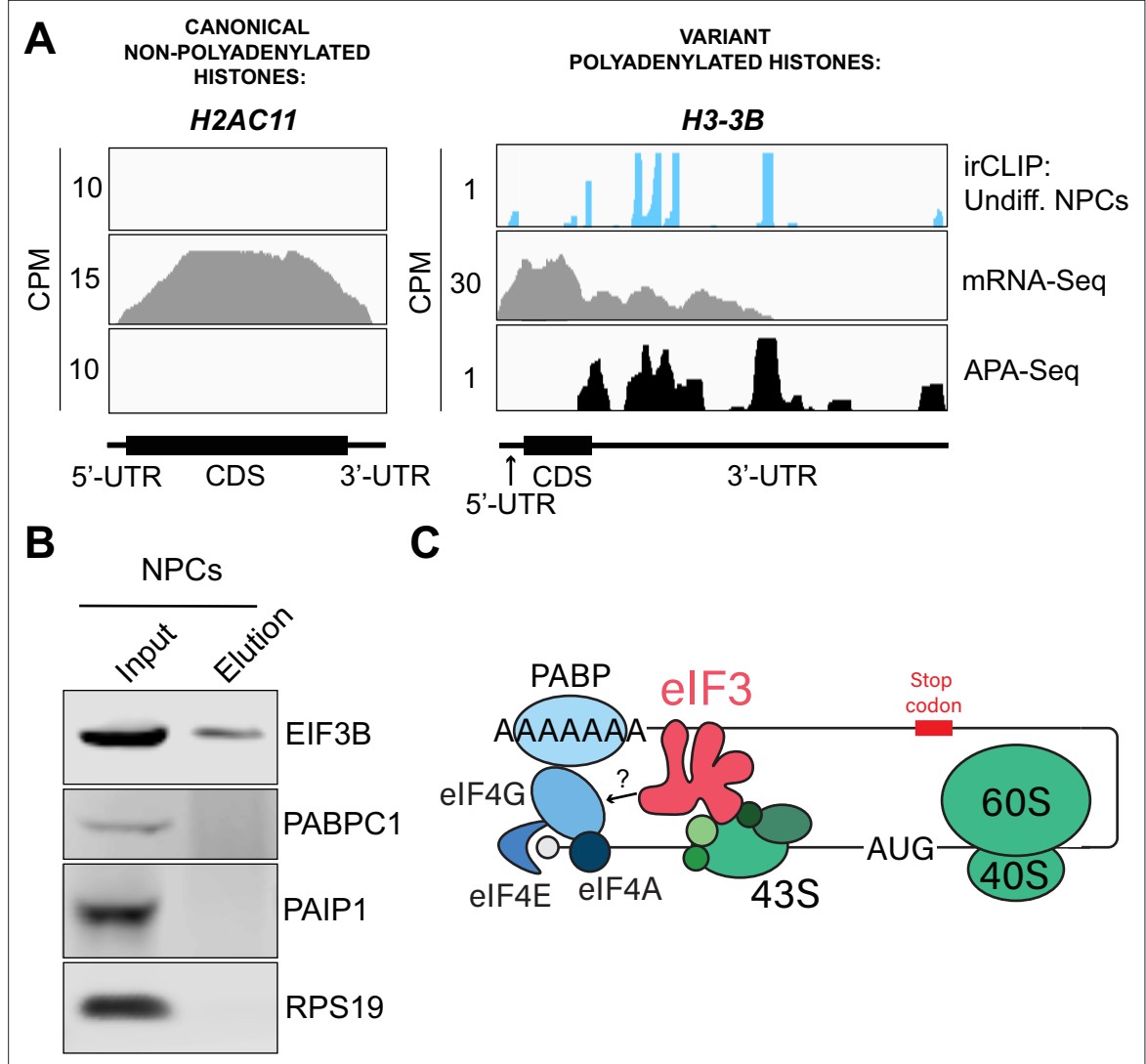

**Figure 4.** Crosslinking of eukaryotic initiation factor 3 (eIF3) to polyadenylated mRNAs and model for mRNA circularization. (**A**) Crosslinking of eIF3 across the canonical non-polyadenylated histone H2AC11 mRNA and the variant polyadenylated histone H3-3B mRNA in undifferentiated neural progenitor cells (NPCs). Polyadenylation sequencing (APA-seq) and mRNA-seq data for these regions are also shown. (**B**) Western blots of EIF3B immunoprecipitations from undifferentiated NPCs. (**C**) Model for eIF3 contribution to mRNA circularization in highly translated transcripts.

The online version of this article includes the following source data and figure supplement(s) for figure 4:

**Source data 1.** Original western blots for *Figure 4B* and *Figure 4—figure supplement 1B*.

**Source data 2.** Original western blots for *Figure 4B* and for *Figure 4—figure supplement 1B*, with experimental conditions indicated.

**Figure supplement 1.** Mechanism of eukaryotic initiation factor 3 (eIF3) binding to mRNA 3'-UTR elements.

**Figure supplement 1—source data 1.** Original western blots in *Figure 4—figure supplement 1C*.

**Figure supplement 1—source data 2.** Original western blots in *Figure 4—figure supplement 1C*, with cells and experimental conditions indicated.

**Figure supplement 2.** Crosslinking of eukaryotic initiation factor 3 (eIF3) to mRNA 3'-UTRs of various lengths in neural progenitor cells (NPCs).

*Figure 4—figure supplement 1A*). We also carried out a global comparison of the eIF3 Quick-irCLIP and APA-Seq crosslinking peaks, and observed that eIF3 crosslinking to the 3'-UTR termini had a positive association with polyadenylation (Methods). These data indicate that eIF3 only interacts with 3'-UTR termini of mRNAs that are polyadenylated, and thus should be in close proximity to the poly(A) tail.

Our observations that eIF3 interacts with 3'-UTR termini in highly translated mRNAs support a model in which these mRNAs are circularized and their 5' and 3' ends are in close proximity. Since we

observe eIF3 crosslinks adjacent to the poly(A) tail, it is conceivable that this eIF3-mediated mRNA circularization is stabilized through interactions between eIF3 and poly(A)-binding proteins. eIF3 has been reported to interact with the poly(A)-binding protein interacting protein 1 (PAIP1) in HeLa cells through the EIF3G subunit to promote closed loop formation (*Martineau et al., 2008*). As in our Quick-irCLIP experiment, we immunoprecipitated assembled eIF3 complexes using an anti-EIF3B antibody from UV-crosslinked undifferentiated NPCs and immunoblotted against PAIP1 and against the major cytoplasmic isoform of the PABPs (PABPC1). Our results indicate that despite eIF3 being in close proximity to the location of the PABPs, these do not elute with it (*Figure 4B*). Immunoprecipitation of eIF3 complexes from UV-crosslinked HEK293T gave the same results (*Figure 4—figure supplement 1B*). It is possible that the eIF3 interactions with poly(A)-binding proteins are transient and eIF3 immunoprecipitation results in the disruption of these complexes. Therefore, to maintain the integrity of these potentially transient interactions, we treated NPCs with the protein-protein cross-linker dithiobis(succinimidyl propionate) (DSP) before cell collection and eIF3 immunoprecipitation. DSP is a thiol-cleavable crosslinker so eluates can be treated with dithiothreitol (DTT) to cleave the disulfide bond in the spacer arm of DSP as a way to separate eIF3 from its polypeptide interactors. Our results show that even when protein-protein transient interactions are covalently linked by DSP treatment, PABPC1 and PAIP1 do not elute with eIF3 in NPCs (*Figure 4—figure supplement 1C*). The same experiments performed in HEK293T gave the same results (*Figure 4—figure supplement 1C*). Overall, these results suggest that eIF3 engagement with 3′-UTR termini in highly translated transcripts requires polyadenylation but is independent of interactions with poly(A)-binding proteins.

## Discussion

Here, we identified the transcripts that interact with eIF3 in hPSC-derived NPCs. NPCs that have undergone differentiation toward forebrain neurons exhibit a global increase in protein synthesis that we identified to occur 2 hr post differentiation treatment. Our Quick-irCLIP experiment shows that eIF3 predominantly crosslinks to 3′-UTRs in NPCs. By performing APA-Seq, we discovered that these eIF3 3′-UTR crosslinks map to 3′-UTR termini of multiple mRNA isoforms, adjacent to poly(A) tails and where the polyadenylation signal is located. Interestingly, we have also identified eIF3 irCLIP peaks within mRNA coding regions such as in *NES* mRNA (*Figure 2—figure supplement 2A*) that colocalize with APA-Seq peaks, indicating that different nestin protein isoforms are expressed in NPCs. Therefore, in addition to providing mRNA isoform-specific expression information, our APA-Seq experiment could potentially be used to investigate the expression of specific protein isoforms in NPCs.

Here, we also observe that early NPC differentiation results in eIF3 engagement with the 3′-UTR terminus of the mRNA encoding the transcription regulator ID2. By ribosome profiling, we determined that this increase in eIF3 engagement on *ID2* mRNA is correlated with an increase in the levels of active transcription and translation of ID2 (*Figure 3*). At a global level, we also observe in both undifferentiated and differentiated NPCs mRNAs with higher levels of eIF3 crosslinks to 3′-UTRs exhibit higher levels of active translation (*Figure 3F* and *Figure 3—figure supplement 4*), suggesting that these two phenomena are correlated. For example, we observe that the highest levels of eIF3 crosslinking at 3′-UTR ends are observed on the most actively translated mRNAs (e.g. *EEF1A* and *ACTB*, *Figure 3E*). Overall, these data suggest that actively translated mRNAs engage eIF3 at their 3′-UTR termini, likely reflecting eIF3's general role in translation. However, the underlying molecular mechanisms of the eIF3 engagement with the 3′-UTR terminus remain to be determined.

Our APA-Seq experiment performed in NPCs maps all polyadenylation events and it infers mRNA expression at isoform level. We found several instances where eIF3 CLIP peaks that colocalize with polyadenylation events have different peak heights with respect to those of the APA-Seq experiment. For instance, our APA-Seq data suggests that the proximal and distal 3′-UTR isoforms of the *S100B* mRNA are expressed at similar levels in NPCs (*Figure 2D*). However, we observe higher eIF3 engagement levels with the distal isoform than with the proximal isoform. Given that our ribosome profiling experiment indicates eIF3 engagement at 3′-UTR termini is observed on actively translated mRNAs, our eIF3 irCLIP and APA-Seq datasets could potentially be used in combination to determine active translation levels at mRNA isoform level, a feature that cannot be determined by ribosome profiling alone since RPF counts cannot be mapped to multiple mRNA isoforms.

eIF3 is an initiation factor that is usually associated with 5′-UTRs, where it organizes interactions between other initiation factors and the 40 S ribosomal subunit during translation initiation. Its

presence at the very end of 3'-UTRs of actively translated mRNAs supports the mRNA circularization model, where the 5' and 3' ends of highly translated mRNAs are proposed to be brought in close proximity to each other by the interactions of multiple proteins that bridge 5'-UTRs with 3'-UTRs (*Wells et al., 1998*). It is possible that our observation of eIF3 at 3'-UTR termini is due to the recycling of eIF3 from 3'-UTRs to 5'-UTRs for successive rounds of translation. However, this would require eIF3 association with translating ribosomes until translation termination and subsequently scanning through the entire length of 3'-UTRs either by direct interactions with mRNAs or through binding with post-termination 40 S ribosomal subunits scanning through 3'-UTRs until the poly(A) tail is encountered. If that were the case one would expect to observe higher eIF3 levels at the ends of short 3'-UTRs than at those of long 3'-UTRs, and we observe no evidence of this (i.e. *Figure 4—figure supplement 2*). Furthermore, recent results indicate that eIF3 associates with elongating ribosomes but it dissociates after ~60 codons from the start codon, before termination (*Lin et al., 2020*). Alternatively, given the large size of eIF3 (800 KDa in humans), it is possible that during mRNA circularization eIF3 interactions with 3'-UTR termini are accomplished by a specific eIF3 module while performing its canonical role as a scaffold protein during translation initiation (*Figure 4C*). In the closed loop model of translation it is proposed that the 5' and 3' ends of translated mRNAs are connected via the multiprotein-RNA interactions composed of the mRNA 5' cap - eIF4E - eIF4G - PABP - poly(A) tail (*Wells et al., 1998*), which are conserved across phylogeny. The disruption of these interactions results in a decrease in translation (*Borman et al., 2000*; *Imataka et al., 1998*; *Kahvejian et al., 2005*; *Tarun and Sachs, 1996*). mRNA circularization promoted by the canonical closed loop model has been observed in vitro by atomic force microscopy (*Wells et al., 1998*). Circular polysomes have also been observed on the rough endoplasmic reticulum (*Christensen et al., 1987*). These studies remain the only structural evidence for the model to date.

Since these studies, additional interactions between proteins regularly associated with 5'-UTRs and proteins commonly associated with 3'-UTRs have been discovered. For example, EIF3H has been shown to interact with METTL3, which binds to *N*6-methyladenosine (m6A) modified sites near the stop codon, an interaction proposed to enhance translation, indicating a role in looping highly translated mRNAs (*Choe et al., 2018*). Another eIF3 subunit, EIF3G, has also been reported to be involved in mRNA circularization. Studies performed in HeLa cells showed EIF3G coimmunoprecipitates with PAIP1, suggesting a role in bridging 5'-UTRs with poly(A) tails (*Martineau et al., 2008*). Here, we immunoprecipitated the eIF3 complex from NPCs and HEK293T using an anti-EIF3B antibody but did not observe PAIP1 or other poly(A)-associated proteins in the eIF3 eluates. It is possible that these divergent results are due to the fact that HeLa extracts present high levels of EIF2α phosphorylation, which are not observed in other cell extracts such as HEK293T (*Aleksashin et al., 2023*), suggesting that protein translation is highly dysregulated in HeLa cells and, therefore, the mechanisms of translation regulation might be significantly different than in other cells. Furthermore, the EIF3G-PAIP1 interaction is reported to be RNA independent, while our observations reveal an interaction of eIF3 with 3'-UTRs of polyadenylated mRNAs, so it is also possible that these are two independent mechanisms. Future studies will be needed to identify the proteins that may contact eIF3 when it resides at the end of the 3'-UTR adjacent to the poly(A) tail.

It is worth noting that the eIF3 3'-UTR termini crosslinking pattern was not observed in the two previous studies that used PAR-CLIP to identify the eIF3 mRNA-binding sites in HEK293T cells (*Lee et al., 2015*) and Jurkat cells (*De Silva et al., 2021*). Quick-irCLIP uses a short UV wavelength (254 nm) to crosslink RNA to proteins, whereas PAR-CLIP is performed at 365 nm and requires the cellular internalization of 4-thiouridine (4SU). These different UV wavelengths lead to different chemical susceptibilities for amino acid-nucleotide crosslinks (*Ascano et al., 2012*). eIF3 is a large multisubunit complex so it is possible that different UV wavelengths capture transcript interactions through different eIF3 modules containing different amino acid propensity to crosslink to mRNAs at the two wavelengths. Using a different UV crosslinking method with respect to the previous work performed in HEK293T and Jurkat is a caveat when it comes to comparative studies such the one reported in *Figure 1—figure supplement 2*, in which eIF3-crosslinked mRNAs in NPCs are nearly disjoint from those crosslinked to eIF3 in HEK293T and Jurkat cells. However, it is striking that we were still able to identify some common 5'-UTR eIF3 binding sites (*Figure 1—figure supplement 3*).

Overall, our results provide new insights into the broad repertoire of roles that eIF3 plays during translation initiation. Besides its canonical role as a scaffold protein for initiation complex assembly

(*Jackson et al., 2010*) and its non-canonical roles in translation activation/repression by direct mRNA interactions at 5'- and 3'-UTRs (*De Silva et al., 2021*; *Lee et al., 2015*), our study reveals a role in which eIF3 interacts with 3'-UTR termini of highly translated mRNAs and suggests eIF3 assists in the communication between 5' and 3' mRNA ends, supporting the mRNA circularization model to promote efficient recycling of ribosomes and translation factors for successive rounds of translation.

## Materials and methods

### Cell lines

Human pluripotent stem cells (WTC-11 cells) and HEK293T cells were obtained from the UC Berkeley Biosciences Cell Culture Facility. The WTC-11 cells were validated using STR, Gband analysis, and markers for stemness by the facility. All cell lines were screened for mycoplasma by the facility using fluorescence microscopy.

### Maintenance and propagation of hPSCs

The surface of all culture ware used for the maintenance and propagation of human pluripotent stem cells (WTC-11 line) was coated with Corning Matrigel (Fisher Scientific) prior to cell seeding. hPSCs were maintained under feeder-free conditions in complete mTeSR Plus medium (Stem Cell Technologies) and passaged using Gentle Cell Dissociation Reagent (Stem Cell Technologies).

### Generation of hPSC-derived NPCs

NPCs were generated from hPSCs using the embryoid body protocol (Stem Cell Technologies) following the manufacturer's protocol. At day 17, hPSC-generated NPCs (passage 1) were maintained and propagated using complete STEMdiff Neural Progenitor Medium (Stem Cell Technologies). Cells from passages 1–6 were cryopreserved.

### Maintenance and propagation of hPSC-derived NPCs

The surface of all cultureware used for the maintenance and propagation of hPSC-derived NPCs was coated with Corning Matrigel (Fisher Scientific) prior to cell seeding. Maintenance and propagation of hPSC-derived NPCs was performed following manufacturer's instructions. Briefly, NPCs were maintained using complete STEMdiff Neural Progenitor Medium (Stem Cell Technologies) for 7–9 d, with medium change every other day. NPCs were passaged using Accutase (Stem Cell Technologies) and seeded onto Matrigel-coated culture ware at an initial concentration of $1.25 \times 10^5$ cells/mL.

### Differentiation treatment of hPSC-derived NPCs and identification of the global increase in protein synthesis

For differentiation studies, hPSC-derived NPCs were seeded at an initial concentration of $5 \times 10^5$ cells/mL in STEMdiff Neural Progenitor Medium (Stem Cell Technologies) and allowed to sit at 37 °C for 48 hr. After 48 hr, the medium was changed to complete STEMdiff Forebrain Neuron Differentiation medium (Stem Cell Technologies) (for differentiated NPCs) or to fresh complete STEMdiff Neural Progenitor Medium (for undifferentiated NPCs) for 2 hr at 37 °C and collected by washing with 1 mL PBS and scraping, followed by brief centrifugation. Cells were stored at –80 °C until further use. For protein synthesis assays, cells were treated with 20 mM puromycin for 15 min at 37 °C prior to collecting, and Western blotting was performed with an anti-puromycin antibody (Abcam, ab315887; 1:1000 dilution) and imaged using Li-COR Odyssey CLx.

### eIF3 immunoprecipitation

Fifty μL of slurry Dynabeads Protein G (Invitrogen) were washed twice with 1 mL PBS +0.01% Tween 20. Then, 25 μL of anti-EIF3B antibody (Bethyl, A301-760A) were added to washed beads and incubated at room temperature for 40 min with mixing by gentle rotation. For IPs with anti-EIF3B antibody performed to look for potential interactions with polyA-binding proteins, the anti-EIF3B antibody was crosslinked to Dynabeads Protein G with (bis(sulfosuccinimidyl)suberate) (BS3) (Thermo Fisher Scientific) following the manufacturer's protocol. For cell lysis, NPCs were lysed using lysis buffer (50 mM Hepes-KOH, pH 7.5, 150 mM KCl, 5 mM $MgCl_2$, 2 mM EDTA, 0.5% Nonidet P-40 alternative, 0.5 mM DTT, 1 Complete EDTA-free Proteinase Inhibitor Cocktail tablet per 10 mL of buffer). For IPs

performed to look for potential EIF3B transient interactions with polyA-binding proteins, prior to collection, NPCs and HEK293T were treated with the protein-protein crosslinker dithiobis[succinimidylpropionate] (DSP) (1 mM) in PBS for 30 min at room temperature, and reaction was quenched by the addition of 10 mM Tris-HCl, pH 7.5 for 15 min at room temperature. Cells were then collected and lysed as described above. The cell pellets resuspended in lysis buffer were incubated on ice for 10 min, passed through a 18 G needle four times, and centrifuged at 13,000 g for 10 min at 4 °C. Twenty µL of lysate was saved as 5% input and the remaining lysate was loaded to washed beads conjugated to anti-eIF3b antibody and incubated at 4 °C for 2 hr with gentle rotation. Next, the beads were washed three times with lysis buffer. Then, eIF3-bound RNAs were eluted from the beads with 50 µL 1 X NuPAGE LDS sample buffer and boiled at 70 °C for 5 min. Samples were loaded onto a 4–12% Bis-Tris gel (Invitrogen), transferred to a nitrocellulose membrane, which was blotted against the desired proteins.

## Quick-irCLIP

Quick-irCLIP was performed as previously described (*Kaczynski et al., 2019*). NPCs subjected to Quick-irCLIP were prepared as follows. NPCs were seeded at an initial concentration of $5 \times 10^5$ cells/mL and incubated at 37 °C for 48 hr in STEMdiff Neural Progenitor Medium. Then, the medium was changed to either fresh STEMdiff Neural Progenitor Medium (undifferentiated cells) or STEMdiff Forebrain Neuron Differentiation medium (differentiated NPCs) and incubated at 37 °C for 2 hr. Cells were then crosslinked by irradiating them on ice with 0.15 J/cm$^2$ of 254 nm UV light. Then, cells were collected by scraping in 1 mL PBS followed by centrifugation at 150×*g* for 5 min and stored at –80 °C until further use. Next, NPC pellets were allowed to thaw on ice for 10 min. In the meantime, 100 µL of slurry Dynabeads (Invitrogen) were washed twice with 1 X PBS +0.01% Tween 20. Then, 25 µL of anti-EIF3B antibody (Bethyl, A301-760A) were added to the washed beads solution and incubated at room temperature for 40 min with gentle rotation. Then, thawed NPCs were lysed by the addition of 1 mL of lysis buffer (50 mM Hepes-KOH, pH 7.5, 150 mM KCl, 5 mM MgCl$_2$, 2 mM EDTA, 0.5% Nonidet P-40 alternative, 0.5 mM DTT, 1 Complete EDTA-free Proteinase Inhibitor Cocktail tablet per 10 mL of buffer) and allowed to sit on ice for 10 min. The resuspended pellet was then passed through a 18 G needle four times, and centrifuged at 13,000×*g* for 10 min at 4 °C. Lysate protein concentration was determined by Bradford assay to determine the amount needed to obtain 0.5 mg of protein. Then, lysate with 0.5 mg per sample was transferred to a new Eppendorf tube. Ten µL of RNase I (Fisher Scientific) (1:1000 dilution) and 2 µL of Turbo DNase (Fisher Scientific) were added to the lysate and allowed to mix at 37 °C for 3 min while shaking at 1100 rpm. We chose a 1:1000 dilution of RNase I because it is sufficient to yield significant RNA smears where the RNA-binding eIF3 subunits EIF3A, EIF3B and EIF3D are located (*Figure 1—figure supplement 5*). Samples were then immediately placed on ice for 3 min followed by centrifugation at 18,000×*g* for 10 min at 4 °C. Supernatants were then transferred to new tubes. At this point, washed beads conjugated to anti-EIF3B antibody were washed twice with 1 X PBS +0.01% Tween 20 and washed once with lysis buffer. Then, RNAse-treated lysates were loaded to conjugated beads and incubated at 4 °C for 2 hr with gentle rotation. Beads were then washed twice with high-salt wash buffer (50 mM Hepes-KOH, pH 7.5, 500 mM KCl, 0.5% Nonidet P-40 alternative, 0.5 mM DTT, 5 mM MgCl$_2$, 1 Complete EDTA-free Proteinase Inhibitor Cocktail tablet per 10 mL of buffer), and then with PNK wash buffer (20 mM Tris-HCl, pH 7.5, 10 mM MgCl$_2$, 0.2% Tween-20). Beads were subsequently resuspended in 20 µL dephosphorylation reaction mix (1 X PNK buffer, pH 6.5, 5 units of T4 polynucleotide kinase (New England Biolabs), 0.5 µL RNaseOut Ribonuclease inhibitor (Fisher Scientific)) and incubated for 20 min at 37 °C while shaking at 1100 rpm. Next, beads were washed once with PNK buffer, then washed once with high-salt wash buffer, and then washed again once with PNK buffer.

The following steps describe the ligation of the adaptor previously conjugated to an IR dye. Briefly, 20 µL of ligation reaction mix was prepared (1 X ligation buffer, 5 units of T4 RNA ligase I (New England Biolabs), 0.5 µL RNaseOut Ribonuclease inhibitor, 1.5 µL of 10 mM infrared adaptor oligonucleotide, 4 µL polyethylene glycol 400). PNK buffer was removed from the beads and then 20 µL of ligation reaction mix were added to the beads and allowed to incubate at 16 °C while shaking at 1100 rpm overnight.

Next, beads were washed once with PNK buffer, twice with high-salt wash buffer, and once more with PNK buffer. Beads were then transferred to new Eppendorf tubes and resuspended in PNK

buffer. Meanwhile, 500 mL of 1 X NuPAGE MOPS-SDS buffer were prepared and 500 µL antioxidant (Fisher Scientific, cat#: NP0005) were added fresh before loading buffer use. PNK buffer was then removed from beads and samples were resuspended with 13 µL nuclease-free water, 2 µL sample reducing agent (Fisher Scientific, cat#: NP0004), and 5 µL 4 X NuPAGE protein loading buffer. Samples were then incubated at 80 °C for 5 min, briefly centrifuged, and loaded onto a 4–12% SDS NuPAGE Bis-tris gel. The gel was run for 50 min at 180 V. Protein-RNA complexes were then transferred to a nitrocellulose membrane for 90 min at 30 V and then were visualized using a near infrared imager (Li-COR Odyssey CLx). The membrane was then put in a light-protected box. A grayscale image of the membrane was printed on acetate film, the membrane was wrapped in a plastic wrap and then the membrane and grayscale image were aligned using the ladder as reference.

Using a razor blade per sample, a box corresponding to the molecular weight range of eIF3 subunits EIF3A, EIF3B, EIF3C, and EIF3D (140–65 kDa) was cut into small pieces and transferred to a new Eppendorf tube. Proteins in the cut box were then digested by adding 10 µL Proteinase K (20 ug/µL) and 200 µL PK buffer (100 mM Tris-HCl, pH 7.4, 50 mM NaCl, 10 mM EDTA) and incubated at 37 °C for 20 min while shaking at 1100 rpm. Then, 200 µL PK-Urea buffer (100 mM Tris-HCl, pH 7.4, 50 mM NaCl, 10 mM EDTA, 7 M urea) and incubated for an additional 20 min while shaking at 37 °C at 1100 rpm. Supernatants were then transferred to phase lock heavy columns along with 400 µL neutral phenol-chloroform and incubated for 5 min at 30 °C while shaking at 1100 rpm. Samples were then centrifuged for 5 min at 18,000×$g$ at room temperature. The aqueous upper layers were transferred to new low-binding 1.5 mL tubes and centrifuged for 1 min at 18,000×$g$ at room temperature. Samples were then transferred to new low-binding 1.5 mL tubes and 0.75 µL glycogen (RNA grade, Fisher Scientific, cat#: FERR0551), 40 µL 3 M sodium acetate (pH 5.5), and 1 mL 100% ethanol were added per sample. Samples were briefly vortexed and precipitated overnight at –20 °C.

The next day, samples were centrifuged for 20 min at 18,000×$g$ at 4 °C, supernatants removed, leaving approximately 50 µL of the RNA pellet. Then, 1 mL of 80% ethanol was added, without overly agitating the pellet. Supernatants were carefully removed and pellets were air-dried at room temperature. Pellets were then resuspended in 7 µL nuclease-free water and transferred to new PCR tubes. Sequencing libraries were created following the protocol described in the SMARTer smRNA-Seq Kit for Illumina user manual (Takara Bio).

## Total RNA cDNA library preparation

RNA samples were extracted from differentiated NPCs or undifferentiated NPCs treated for 2 hr with STEMdiff Forebrain Neuron Differentiation medium (Stem Cell Technologies) for differentiated NPCs, or with fresh complete STEMdiff Neural Progenitor Medium for undifferentiated NPCs, using Trizol reagent (Thermo Fisher). cDNA libraries were prepared using NEBNext ultra II Directional RNA Library Prep Kit for Illumina (NEBNext rRNA Depletion Kit v2).

## Quick-irCLIP computational analysis

Quick-irCLIP and Total RNA cDNA libraries were sequenced on an Illumina NovaSeq S1 150PE platform. Cutadapt (*Martin, 2011*) (version 3.5), with a minimum length of 20 nucleotides, was used to remove 3′ adapter sequences. Strand-specific alignments were generated using HISAT v.2.2.1 (*Kim et al., 2019*) and hg38. Duplicate reads were removed with Picard (v2.21.9, https://broadinstitute.github.io/picard/; *Broad Institute, 2020*) MarkDuplicates, followed by peak calling with HOMER (v4.11) findPeaks and a FDR cutoff of 0.01 (*Heinz et al., 2010*). Bedtools intersect (v2.29.2) was used to identify statistically significant peaks (BH-adjusted p-value <1.0e-04) present in all replicates, then motifs were generated with HOMER findMotifsGenome. Peak statistics were tabulated using custom Python scripts. For coverage graphs, bams were CPM (Counts Per Million) normalized and converted to BigBiwgs with Deeptools (v3.5.1) (*Ramírez et al., 2016*) then visualized with IGV. Figures without irCLIP and APA-seq replicate tracks represent the union of all replicate reads, prepared with Samtools (v1.17) (*Danecek et al., 2021*) merge and CPM normalization. To assess the consistency of irCLIP bioreplicates (n=3), the Spearman Correlation coefficient was calculated using dedpulicated, CPM normalized BigWigs and DeepTools plotCorrelation. The average correlation coefficients were 0.7855 (standard deviation: 0.0092) for differentiated NPCs and 0.834 (standard deviation: 0.0058) for undifferentiated NPCs, indicating high reproducibility (*Figure 1—figure supplement 6*). Additionally, to investigate the relationship between eIF3 3'-UTR crosslinking and translation (*Figure 3F*,

*Figure 3—figure supplement 4A*), the Spearman correlation coefficient was calculated between each transcript's average irCLIP RPKM and mean TE across replicates (n=3) for differentiated and undifferentiated cells. The Spearman correlation was also determined for irCLIP RPKM and average RPF coverage, normalized for coding sequence (CDS) length. In the case of TE values, outliers exceeding the TE's 99th percentile were removed, to make the log-log plots more readable.

## Comparison of eIF3 Quick-irCLIP results in NPCs with eIF3 PAR-CLIP results in Jurkat and HEK293T cells

Since the eIF3 PAR-CLIP experiments done in Jurkat and HEK293T cells were performed to identify the transcripts that interact with individual eIF3 subunits, first we obtained the RNAs that are common interactors of eIF3 subunits EIF3A (328 RNA-EIF3A reported clusters), EIF3B (264 RNA-EIF3B reported clusters), and d (356 RNA-EIF3D reported clusters) in HEK293T (*Lee et al., 2015*). The reason we only picked these subunits for this comparison study is because in our Quick-irCLIP experiment we only isolated the RNAs bound to these subunits. This rendered a list of 175 RNA-eIF3 common clusters between subunits EIF3A, EIF3B, and EIF3D in HEK293T. We then picked the top 400 transcripts that interact with activated Jurkat cells (*De Silva et al., 2021*) and performed the same analysis to find the common RNAs that bind to subunits EIF3A, EIF3B, and EIF3D, giving a total of 209 transcripts. Next, we identified the common transcripts between the two studies by comparing the 175 common in HEK293T with the 209 common in activated Jurkat cells, giving a total of 27 transcripts. Finally, we picked the top 210 transcripts (common in the three replicates) identified by eIF3 Quick-irCLIP in differentiated NPCs and compared them to the 175 common RNAs in HEK293T cells (resulting in eight NPC-HEK293T common RNAs) and to the 209 common RNAs in activated Jurkat cells (resulting in 12 NPC-Jurkat common RNAs). Performing the same analysis with the top 210 common transcripts that bind eIF3 in undifferentiated NPCs gave the same results.

## Alternative polyadenylation (APA)-seq

Total RNA was extracted using Trizol Reagent from NPCs that were seeded at an initial concentration of $5 \times 10^5$ cells/mL and incubated at 37 °C for 48 hr in STEMdiff Neural Progenitor Medium. After 48 hr, the media was changed to fresh STEMdiff Neural Progenitor Medium and NPCs were incubated 37 °C for 2 hr and collected. Total RNA was then extracted (Trizol reagent) from NPCs. 3' mRNA-Seq libraries were prepared from 500 ng total RNA using QuantSeq REV kits (Lexogen) according to the manufacturer's protocol.

## APA-seq computational analysis

APA-Seq and Total RNA cDNA libraries were sequenced on an Illumina NovaSeq S1 150PE platform. Cutadapt (version 3.5) (*Martin, 2011*) with a minimum length of 20 nucleotides was used to remove 3' adapter sequences. RNA-seq reads were aligned using Hisat2 v.2.2.1 (*Kim et al., 2019*) and mapped to the hg38 reference genome. SAMtools (*Danecek et al., 2021*) was used to remove PCR duplicates and filter uniquely mapped reads. Peaks were called with HOMER (v4.11) (*Heinz et al., 2010*) findPeaks at an FDR cutoff of 0.01.

## Statistical analysis of irCLIP eIF3 crosslinking and APA-seq

We performed a statistical analysis to assess the dependency of eIF3 3'-UTR engagement on polyadenylation. For precision, non-overlapping 3'-UTRs were selected from transcripts expressed in the mRNA-Seq dataset (TPM >1). APA-Seq and irCLIP peaks, consistent in all three biological replicates within these 3'-UTRs, were analyzed using a Fisher's Exact test. The analysis yielded a p-value of $9.993 \times 10^{-43}$ and an odds ratio of 5.616, indicating a significant positive association between eIF3 3'-UTR binding and polyadenylation. To address the fact that APA-Seq and irCLIP were performed separately and each method cannot be used to distinguish between distinct mRNA isoforms, we focused the analysis on reproducible peaks within non-overlapping 3'-UTRs of expressed transcripts.

## Ribosome profiling

### Preparation of NPC lysates

Ribosome profiling was performed as previously described (*Ferguson et al., 2023*; *McGlincy and Ingolia, 2017*), with the following modifications. Briefly, NPCs were seeded in four Matrigel-coated

15 cm dishes per condition and allowed to reach 70–80% confluency in STEMdiff Neural Progenitor Medium (Stem Cell Technologies). Then, they were treated for 2 hr with STEMdiff Forebrain Neuron Differentiation medium for differentiated NPCs, or STEMdiff Neural Progenitor Medium for undifferentiated NPCs. NPCs were treated with 100 µg/mL cycloheximide for 5 min prior to collection and washed with ice-cold PBS containing 100 µg/mL cycloheximide. PBS was discarded and 1.2 mL Lysis Buffer (20 mM Tris-HCl, pH 7.4, 150 mM NaCl, 5 mM MgCl2, 1 mM DTT, 100 µg/mL cycloheximide, 1 % v/v Triton X-100, 25 U/mL Turbo DNase I) was added and cells were scraped, transferred to an Eppendorf tube, and allowed to sit on ice for 15 min. Cells were then passed 10 times through a 26 gauge needle and lysate was clarified by centrifugation (10 min, 20,000×$g$, 4 °C). Supernatants were recovered and stored at –80°C.

## RNA quantification of NPC lysates
RNA concentration of lysates diluted in 1 X TE buffer was quantified by using Quant-iT RiboGreen RNA kit (ThermoFisher) through direct comparison with a 0.0–1.0 ng/µL rRNA standard curve (Thermo Fisher).

## P1 nuclease treatment of NPC lysates and ribosome pelleting
The pH of lysates was adjusted to 6.5 for optimal P1 nuclease activity by adding RNase-free 300 mM Bis-Tris pH 6.0 per 100 µL of cell lysate. Then, 450 Units of P1 Nuclease (100 U/µL, New England Biolabs) were added per 30 µg of RNA and lysates were incubated at 37°C for 1 hr with gentle rotation. In the meantime, sucrose density gradients were prepared by dissolving 1 M D-sucrose in 10 mL Polysome Buffer (20 mM Tris-HCl, pH 7.4, 150 mM NaCl, 5 mM MgCl$_2$, 1 mM DTT, 100 µg/mL cycloheximide) and 900 µL were added to 13×56 mm polycarbonate thick wall tube (Beckman Coulter). Then, 300 µL of digested lysates were carefully added to the top of the sucrose gradient and centrifuged at 100,000 RPM for 1 hr at 4°C in a pre-chilled 4°C TLA 100.3 rotor. The sucrose cushion was then aspirated and 30 µL nuclease-free water were added to the ribosome pellet, allowing it to sit on ice for 10 min. The ribosome pellet was disrupted by pipetting and 300 µL TRIzol (Thermo Fisher) were added and the entire volume was transferred to a pre-chilled tube, vigorously vortexed and stored at –80°C.

## mirRICH small RNA enrichment from pelleted ribosomes
A modified TRIzol (Thermo Fisher) RNA extraction (*Choi et al., 2018*) was performed by first adding chloroform, vortexing, centrifuging, and isolating the aqueous phase as described by the manufacturer. RNA was precipitated by adding 100% isopropanol and incubating on ice for 15 min. RNA was pelleted by centrifugation at max speed for 15 min at 4°C. Supernatant was carefully removed and pellets were allowed to air-dry at room temperature for 2 hrs. Dried pellets were then resuspended in 10 µL nuclease-free water, vortexed, and briefly centrifuged and incubated at room temperature for 5 min. Resuspended pellets were vortexed and briefly centrifuged again and 9 µL of the eluate liquid was removed and purified by RNA & Clean Concentrator-5, eluted in 25 µL nuclease-free water, and quantified by Nanodrop.

## cDNA size selection, elution, and quantification
The protocol from *Ferguson et al., 2023* was followed for the OTTR reaction. OTTR reaction products were resuspended in bromophenol blue formamide loading dye. OTTR cDNAs were resolved by electrophoresis in 0.6 X TBE 8% Urea-PAGE and visualized by fluorescence using the 5' Cy5-dye. Gel excision and elution was performed as described (*Ferguson et al., 2023*). To quantify cDNA by qPCR and determine the number of PCR cycles needed for library amplification, we followed a previously described method (*McGlincy and Ingolia, 2017*).

## Ribosome profiling computational analysis
Ribosome profiling and Total RNA cDNA libraries were sequenced on an Illumina NovaSeq 150PE platform. The adapters were trimmed, the unique molecular identifier (UMI) was appended to the FASTQ entry header, and reads shorter than 20 were excluded using Cutadapt, as described (*Ferguson et al., 2023*):

```
cutadapt -m 2 a GATCGGAAGAGCACACGTCTGAACTCCAGTCAC $tmpvar.fastq |
cutadapt -u 7 --rename='{id} UMI={cut_prefix}' - | cutadapt -u −1
--rename='{id}_{comment}_primer={cut_suffix}' - | cutadapt -m 20 -q 10 -
```

Human references described here were used as in *Ferguson et al., 2023*. Briefly, human rRNA sequences NR_023379.1, NR_146151.1, NR_146144.1, NR_145819.1, NR_146117.1, NR_003285.3, NR_003286.4, X12811.1, and NR_003287.4 from NCBI and ENST00000389680.2 and ENST00000387347.2 from Ensembl were used as the rRNA reference. The human tRNA sequences and references were prepared as described (*Holmes et al., 2022*). ncRNA from Ensembl and lncRNA from Gencode were concatenated. The primary assembly of the human gDNA from NCBI, GRCh38, was used. 18,640 NCBI RefSeq MANE v0.95 transcripts were used, and the 5'-UTR, CDS, and 3'-UTR lengths were parsed from the GenBank entries using a custom script. The 13 mitochondrial mRNA sequences from Ensembl were also included in the mRNA reference, but typically excluded from analysis.

Contaminating reads were removed with a sequential analysis pipeline as described (*Ferguson et al., 2023*). Briefly, the trimmed reads were first mapped to the rRNA reference (bowtie -v 3 a --best --norc), then tRNA reference by tRAX (*Holmes et al., 2022*), then ncRNA reference (bowtie -v 3 a --best --norc), and then mtDNA reference (bowtie -v 2 a --best --norc). The remaining reads were then mapped to the mRNA references (bowtie -v 2 m 200 a --norc) and sorted (samtools sort -n). Remaining reads were aligned to the gDNA reference (bowtie -v 2 a --best) before mapping unaligned reads to a reference of size-selecting oligos and adapter oligos (bowtie -v 2 a --best).

To measure CDS occupancy we used Bedtools intersect (v2.25.0) to exclude reads which aligned outside of the +15th to −10th codon of the CDS, counted reads using RSEM (rsem-calculate-expression --alignments --strandedness forward --seed-length 20 --sampling-for-bam). Gene counts were then normalized and analyzed by DESeq2 (*Love et al., 2014*). Three replicates were produced per condition. DESeq2 normalized (median-of-ratios) gene-level counts for each replicated condition were averaged. mRNA sequencing libraries were processed by an identical pipeline and normalized and analyzed by DESeq2. To compare ribosome profiling and irCLIP datasets, the irCLIP reads were re-processed by the same pipeline for ribosome profiling libraries, but only the irCLIP read counts to the 3'-UTR region of genes were analyzed by DESeq2.

For initiation, termination, and solitary codon occupancy profiles, only genes with ≥50 nt of annotated 5'-UTR, ≥450 nt of annotated CDS, ≥50 nt of annotated 3'-UTR, and ≥1 reads per codon were considered. Alignments were counted based on read length and either 5' or 3' position relative to the first base of a codon of interest (e.g. start codon), and rescaled by average number of reads per codon for a given gene. Rescaled counts for each read length and relative position were summed and divided by the number of genes under consideration. For visualization, read lengths were binned (e.g. read lengths of 58, 59, and 60 nt were binned to 58–60 nt) and summed by rescaled counts. A-site codon offsets for P1 nuclease footprints were assigned based on the frame of the 5' end of the read, as described in *Ferguson et al., 2023*.

## Acknowledgements

We thank Dr. Angélica M González-Sánchez and Dr. Yekaterina Shulgina for helpful comments on the manuscript, PM. Quan Mai for helpful discussions, and Dr. Nikolay A Aleksashin for providing us with the human eIF3 representation shown in *Figure 1A*. We thank K Collins and members of the Collins lab for OTTR library generation reagents. SMF was a Postdoctoral CIRM Scholar of the California Institute for Regenerative Medicine program EDUC4-12790. This work was also supported by the National Institutes of Health (NIH) grants R01-GM065050 and R35-GM148352 (to JHDC), and R01-GM139008 (to NTI). LF was supported by a Bakar Fellows Program Award and National Institutes of Health (NIH) DP1 HL156819 (to K Collins).

## Additional information

### Funding

| Funder | Grant reference number | Author |
| --- | --- | --- |
| California Institute for Regenerative Medicine | EDUC4-12790 | Santi Mestre-Fos |
| National Institutes of Health | R01-GM065050 | Jamie HD Cate |
| National Institutes of Health | R35-GM148352 | Jamie HD Cate |
| National Institutes of Health | R01-GM139008 | Nicholas T Ingolia |
| National Institutes of Health | DP1-HL156819 | Lucas Ferguson |
| Bakar Fellows Program | | Lucas Ferguson |

The funders had no role in study design, data collection and interpretation, or the decision to submit the work for publication.

### Author contributions

Santi Mestre-Fos, Conceptualization, Data curation, Formal analysis, Validation, Investigation, Visualization, Methodology, Writing – original draft, Writing – review and editing; Lucas Ferguson, Data curation, Formal analysis, Investigation, Visualization, Methodology, Writing – review and editing; Marena I Trinidad, Data curation, Formal analysis, Validation, Investigation, Visualization, Methodology, Writing – original draft, Writing – review and editing; Nicholas T Ingolia, Supervision, Funding acquisition, Methodology, Writing – review and editing; Jamie HD Cate, Conceptualization, Data curation, Formal analysis, Supervision, Funding acquisition, Investigation, Visualization, Methodology, Writing – original draft, Project administration, Writing – review and editing

### Author ORCIDs

Santi Mestre-Fos ⓘ http://orcid.org/0000-0002-1355-2344
Nicholas T Ingolia ⓘ https://orcid.org/0000-0002-3395-1545
Jamie HD Cate ⓘ https://orcid.org/0000-0001-5965-7902

Joint public review: https://doi.org/10.7554/eLife.102977.3.sa1
Author response https://doi.org/10.7554/eLife.102977.3.sa2

## Additional files

### Supplementary files

Supplementary file 1. Subunits of eukaryotic initiation factor 3 (eIF3) and proteins pulled down by anti-EIF3B IP. The IP was performed using lysates from undifferentiated neural progenitor cells (NPCs). Given are protein names, sequence coverage, and number of peptides detected.

Supplementary file 2. Annotation of clusters from Quick-irCLIP of eukaryotic initiation factor 3 (eIF3) to RNAs in undifferentiated neural progenitor cells (NPCs). Three different filters for conflicts in gene annotation are presented. Filter 0 prioritizes 'mRNA' as an annotation. Filter 1 prioritizes the 3'-UTR in mRNA annotations. Filter 2 prioritizes the 5'-UTR in mRNA annotations. *Figures 1 and 2C* present results using Filter 1.

Supplementary file 3. Annotation of clusters from Quick-irCLIP of eukaryotic initiation factor 3 (eIF3) to RNAs in differentiated neural progenitor cells (NPCs). Three different filters for conflicts in gene annotation are presented. Filter 0 prioritizes 'mRNA' as an annotation. Filter 1 prioritizes the 3'-UTR in mRNA annotations. Filter 2 prioritizes the 5'-UTR in mRNA annotations. *Figures 1 and 2C* present results using Filter 1.

Supplementary file 4. Ribosome profiling and translation efficiency in undifferentiated and differentiated neural progenitor cells (NPCs). DESeq2 comparisons of ribosome protected fragments

(rpf), mRNAs (rna), translation efficiency (te) are given for MANE transcripts.

Supplementary file 5. Normalized mRNA counts from DESeq2. Counts are given for all NPC replicates, using MANE transcripts.

Supplementary file 6. Normalized monosome and disome ribosome protected fragments from DESeq2. Counts are given for all neural progenitor cells (NPC) replicates, using MANE transcripts.

Supplementary file 7. List of antibodies used in this study.

MDAR checklist

## Data availability

Data underlying this article are publicly available. irCLIP-seq and RNA-seq data are deposited under GEO accession GSE246727. Sequencing data from the APA-seq experiment are available as GEO accession GSE246786. Ribosome profiling data are accessible as SRA project ID PRJNA1029246.

The following datasets were generated:

| Author(s) | Year | Dataset title | Dataset URL | Database and Identifier |
|---|---|---|---|---|
| Mestre-Fos S, Ferguson L, Trinidad M, Ingolia NT, Cate JH | 2023 | eIF3 Engages with 3'-UTR Termini of Highly Translated mRNAs in Neural Progenitor Cells [Quick-irCLIP] | https://www.ncbi.nlm.nih.gov/geo/query/acc.cgi?acc=GSE246727 | NCBI Gene Expression Omnibus, GSE246727 |
| Mestre-Fos S, Ferguson L, Trinidad M, Ingolia NT, Cate JH | 2023 | eIF3 Engages with 3'-UTR Termini of Highly Translated mRNAs in Neural Progenitor Cells [APA-Seq] | https://www.ncbi.nlm.nih.gov/geo/query/acc.cgi?acc=GSE246786 | NCBI Gene Expression Omnibus, GSE246786 |
| Mestre-Fos S, Ferguson L, Trinidad M, Ingolia NT, Cate JH | 2023 | eIF3 Engages with 3'-UTR Termini of Highly Translated mRNAs in Neural Progenitor Cells | https://www.ncbi.nlm.nih.gov/sra/?term=PRJNA1029246 | NCBI Sequence Read Archive, PRJNA1029246 |

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
