## [Editor Report · eLife Assessment]

This **valuable** study shows previously unappreciated binding of the eukaryotic translation initiation factor 3 (eIF3) to the poly(A) tail proximal portion of 3' untranslated regions (UTRs) of mRNAs that are efficiently translated in neuronal progenitors. The authors' conclusions are supported by **solid** experimental evidence which is based on several orthogonal systems biology approaches. This article is of considerable interest to the broad spectrum of biomedical researchers interested in studying post-transcriptional regulation of gene expression.

---

## [Referee Report · Joint public review]

Reviewers thought that the authors addressed some, but not all the concerns raised in the previous round of a review.

Strengths: The authors employed a battery of next-generation sequencing and crosslinking techniques (e.g., Quick-irCLIP, APA-Seq, and Ribo-Seq) to describe a previously unappreciated binding of eIF3 to the 3'UTRs of the mRNAs. It is also shown that eIF3:3'UTR binding occurs in the vicinity of poly(A) tail of mRNAs that are actively translated in neuronal progenitor cells derived from human pluripotent stem cells. Collectively, these findings provide evidence for the role of eIF3 in regulating translation from the 3'UTR end of the mRNA.

Weaknesses: In addition to these clear strengths of the article, some weaknesses were observed pertinent to the lack of mechanistic data. It was therefore thought that the experiments aiming to dissect the mechanisms of eIF3 binding to 3'UTRs and their impact on translation warrant future studies. Finally, establishing the impact of the proposed eIF3:3'UTR binding mechanism of translational regulation on cellular fate is required to further support the biological importance of the observed phenomena. It was found that this should also be addressed in the follow up studies.

---

## [Author Response]

The following is the authors’ response to the current reviews.

The authors agree with the reviewers that future studies are needed to dissect the mechanisms of eIF3 binding to 3'UTRs and their impact on translation, and the impact of this binding on cellular fate.

The following is the authors’ response to the original reviews.

**eLife Assessment**
This valuable study reveals extensive binding of eukaryotic translation initiation factor 3 (eIF3) to the 3' untranslated regions (UTRs) of efficiently translated mRNAs in human pluripotent stem cell-derived neuronal progenitor cells. The authors provide solid evidence to support their conclusions, although this study may be enhanced by addressing potential biases of techniques employed to study eIF3:mRNA binding and providing additional mechanistic detail. This work will be of significant interest to researchers exploring post-transcriptional regulation of gene expression, including cellular, molecular, and developmental biologists, as well as biochemists.

We thank the reviewers for their positive views of the results we present, along with the constructive feedback regarding the strengths and weaknesses of our manuscript, with which we generally agree. We acknowledge our results will require a deeper exploration of the molecular mechanisms behind eIF3 interactions with 3'-UTR termini and experiments to identify the molecular partners involved. Additionally, given that NPC differentiation toward mature neurons is a process that takes around 3 weeks, we recognize the importance of examining eIF3-mRNA interactions in NPCs that have undergone differentiation over longer periods than the 2-hr time point selected in this study. Finally, considering the molecular complexity of the 13subunit human eIF3, we agree that a direct comparison between Quick-irCLIP and PAR-CLIP will be highly beneficial and will determine whether different UV crosslinking wavelengths report on different eIF3 molecular interactions. Additional comments are given below to the identified weaknesses.

**Public Reviews:**

**Reviewer #1 (Public review):**
Summary:The authors perform irCLIP of neuronal progenitor cells to profile eIF3-RNA interactions upon short-term neuronal differentiation. The data shows that eIF3 mostly interacts with 3'-UTRs - specifically, the poly-A signal. There appears to be a general correlation between eIF3 binding to 3'-UTRs and ribosome occupancy, which might suggest that eIF3 binding promotes proteinStrengths:The study provides a wealth of new data on eIF3-mRNA interactions and points to the potential new concept that eIF3-mRNA interactions are polyadenylation-dependent and correlate with ribosome occupancy.Weaknesses:(1) A main limitation is the correlative nature of the study. Whereas the evidence that eIF3 interacts with 3-UTRs is solid, the biological role of the interactions remains entirely unknown. Similarly, the claim that eIF3 interactions with 3'-UTR termini require polyadenylation but are independent of poly(A) binding proteins lacks support as it solely relies on the absence of observable eIF3 binding to poly-A (-) histone mRNAs and a seeming failure to detect PABP binding to eIF3 by co-immunoprecipitation and Western blotting. In contrast, LC-MS data in Supplementary File 1 show ready co-purification of eIF3 with PABP.

We agree the molecular mechanisms underlying the crosslinking between eIF3 and the end of mRNA 3’-UTRs remains to be determined. We also agree that the lack of interaction seen between eIF3 and PABP in Westerns, even from HEK293T cells, is a puzzle. The low sequence coverage in the LC-MS data gave us pause about making a strong statement that these represent direct eIF3 interactions, given the similar background levels of some ribosomal proteins.

(2) Another question concerns the relevance of the cellular model studied. irCLIP is performed on neuronal progenitor cells subjected to neuronal induction for 2 hours. This short-term induction leads to a very modest - perhaps 10% - and very transient 1-hour-long increase in translation, although this is not carefully quantified. The cellular phenotype also does not appear to change and calling the cells treated with differentiation media for 2 hours "differentiated NPCs" seems a bit misleading. Perhaps unsurprisingly, the minor "burst" of translation coincides with minor effects on eIF3-mRNA interactions most of which seem to be driven by mRNA levels. Based on the ~15-fold increase in ID2 mRNA coinciding with a ~5-fold increase in ribosome occupancy (RPF), ID2 TE actually goes down upon neuronal induction.

We agree that it will be interesting to look at eIF3-mRNA interactions at longer time points after induction of NPC differentiation. However, the pattern of eIF3 crosslinking to the end of 3’-UTRs occurs in both time points reported here, which is likely to be the more general finding in what we present.

(3) The overlap in eIF3-mRNA interactions identified here and in the authors' previous reports is minimal. Some of the discrepancies may be related to the not well-justified approach for filtering data prior to assessing overlap. Still, the fundamentally different binding patterns - eIF3 mostly interacting with 5'-UTRs in the authors' previous report and other studies versus the strong preference for 3'-UTRs shown here - are striking. In the Discussion, it is speculated that the different methods used - PAR-CLIP versus irCLIP - lead to these fundamental differences. Unfortunately, this is not supported by any data, even though it would be very important for the translation field to learn whether different CLIP methodologies assess very different aspects of eIF3-mRNA interactions.

We agree the more interesting aspect of what we observe is the difference in location of eIF3 crosslinking, i.e. the end of 3’-UTRs rather than 5’-UTRs or the pan-mRNA pattern we observed in T cells. The reviewer is right that it will be important in the future to compare PAR-CLIP and Quick-irCLIP side-by-side to begin to unravel the differences we observe with the two approaches.

**Reviewer #2 (Public review):**
Summary:The paper documents the role of eIF3 in translational control during neural progenitor cell (NPC) differentiation. eIF3 predominantly binds to the 3' UTR termini of mRNAs during NPC differentiation, adjacent to the poly(A) tails, and is associated with efficiently translated mRNAs, indicating a role for eIF3 in promoting translation.Strengths:The manuscript is strong in addressing molecular mechanisms by using a combination of nextgeneration sequencing and crosslinking techniques, thus providing a comprehensive dataset that supports the authors' claims. The manuscript is methodologically sound, with clear experimental designs.Weaknesses:(1) The study could benefit from further exploration into the molecular mechanisms by which eIF3 interacts with 3' UTR termini. While the correlation between eIF3 binding and high translation levels is established, the functionality of these interactions needs validation. The authors should consider including experiments that test whether eIF3 binding sites are necessary for increased translation efficiency using reporter constructs.

We agree with the reviewer that the molecular mechanism by which eIF3 interacts with the 3’UTR termini remains unclear, along with its biological significance, i.e. how it contributes to translation levels. We think it could be useful to try reporters in, perhaps, HEK293T cells in the future to probe the mechanism in more detail.

(2) The authors mention that the eIF3 3' UTR termini crosslinking pattern observed in their study was not reported in previous PAR-CLIP studies performed in HEK293T cells (Lee et al., 2015) and Jurkat cells (De Silva et al., 2021). They attribute this difference to the different UV wavelengths used in Quick-irCLIP (254 nm) and PAR-CLIP (365 nm with 4-thiouridine). While the explanation is plausible, it remains a caveat that different UV crosslinking methods may capture different eIF3 modules or binding sites, depending on the chemical propensities of the amino acid-nucleotide crosslinks at each wavelength. Without addressing this caveat in more detail, the authors cannot generalize their findings, and thus, the title of the paper, which suggests a broad role for eIF3, may be misleading. Previous studies have pointed to an enrichment of eIF3 binding at the 5' UTRs, and the divergence in results between studies needs to be more explicitly acknowledged.

We agree with the reviewer that the two methods of crosslinking will require a more detailed head-to-head comparison in the future. However, we do think the title is justified by the fact that we see crosslinking to the termini of 3’-UTRs across thousands of transcripts in each condition. Furthermore, the 3’-UTR crosslinking is enriched on mRNAs with higher ribosome protected fragment counts (RPF) in differentiated cells, Figure 3F.

(3) While the manuscript concludes that eIF3's interaction with 3' UTR termini is independent of poly(A)-binding proteins, transient or indirect interactions should be tested using assays such as PLA (Proximity Ligation Assay), which could provide more insights.

This is a good idea, but would require a substantial effort better suited to a future publication. We think our observations are interesting enough to the field to stimulate future experimentation that we may or may not be most capable of doing in our lab.

**Reviewer #3 (Public review):**
Summary:In this manuscript by Mestre-Fos and colleagues, authors have analyzed the involvement of eIF3 binding to mRNA during differentiation of neural progenitor cells (NPC). The authors bring a lot of interesting observations leading to a novel function for eIF3 at the 3'UTR.During the translational burst that occurs during NPC differentiation, analysis of eIF3-associated mRNA by Quick-irCLIP reveals the unexpected binding of this initiation factor at the 3'UTR of most mRNA. Further analysis of alternative polyadenylation by APAseq highlights the close proximity of the eIF3-crosslinking position and the poly(A) tail. Furthermore, this interaction is not detected in Poly(A)-less transcripts. Using Riboseq, the authors then attempted to correlate eIF3 binding with the translation efficacy of mRNA, which would suggest a common mechanism of translational control in these cells. These observations indicate that eIF3-binding at the 3'UTR of mRNA, near the poly(A) tail, may participate to the closed-loop model of mRNA translation, bridging 5' and 3', and allowing ribosomes recycling. However, authors failed to detect interactions of eIF3, with either PABP or Paip1 or 40S subunit proteins, which is quite unexpected.Strength:The well-written manuscript presents an attractive concept regarding the mechanism of eIF3 function at the 3'UTR. Most mRNA in NPC seems to have eIF3 binding at the 3'UTR and only a few at the 5'end where it's commonly thought to bind. In a previous study from the Cate lab, eIF3 was reported to bind to a small region of the 3'UTR of the TCRA and TCRB mRNA, which was responsible for their specific translational stimulation, during T cell activation. Surprisingly in this study, the eIF3 association with mRNA occurs near polyadenylation signals in NPC, independently of cell differentiation status. This compelling evidence suggests a general mechanism of translation control by eIF3 in NPC. This observation brings back the old concept of mRNA circularization with new arguments, independent of PABP and eIF4G interaction. Finally, the discussion adequately describes the potential technical limitations of the present study compared to previous ones by the same group, due to the use of Quick-irCLIP as opposed to the PAR-CLIP/thiouridine.Weaknesses:(1) These data were obtained from an unusual cell type, limiting the generalizability of the model.

We agree that unraveling the mechanism employed by eIF3 at the mRNA 3’-UTR termini might be better studied in a stable cell line rather than in primary cells.

(2) This study lacks a clear explanation for the increased translation associated with NPC differentiation, as eIF3 binding is observed in both differentiated and undifferentiated NPC. For example, I find a kind of inconsistency between changes in Riboseq density (Figure 3B) and changes in protein synthesis (Figure 1D). Thus, the title overstates a modest correlation between eIF3 binding and important changes in protein synthesis.

We thank the reviewer for this question. Riboseq data and RNASeq data are not on absolute scales when comparing across cell conditions. They are normalized internally, so increases in for example RPF in Figure 3B are relative to the bulk RPF in a given condition. By contrast, the changes in protein synthesis measured in Figure 1D is closer to an absolute measure of protein synthesis.

(3) This is illustrated by the candidate selection that supports this demonstration. Looking at Figure 3B, ID2, and SNAT2 mRNA are not part of the High TE transcripts (in red). In contrast, the increase in mRNA abundance could explain a proportionally increased association with eIF3 as well as with ribosomes. The example of increased protein abundance of these best candidates is overall weak and uncertain.

We agree that using TE as the criterion for defining increased eIF3 association would not be correct. By “highly translated” we only mean to convey the extent of protein synthesis, i.e. increases in ribosome protected fragments (RPF), rather than the translational efficiency.

(4) Despite several attempts (chemical and UV cross-linking) to identify eIF3 partners in NPC such as PABP, PAIP1, or proteins from the 40S, the authors could not provide any evidence for such a mechanism consistent with the closed-loop model. Overall, this rather descriptive study lacks mechanistic insight (eIF3 binding partners).

We agree that it will be important to identify the molecular mechanism used by eIF3 to engage the termini of mRNA 3’-UTRs. Nevertheless, the identification of eIF3 crosslinking to that location in mRNAs is new, and we think will stimulate new experiments in the field.

(5) Finally, the authors suspect a potential impact of technical improvement provided by QuickirCLIP, that could have been addressed rather than discussed.

We agree a side-by-side comparison of eIF3 crosslinks captured by PAR-CLIP versus QuickirCLIP will be an important experiment to do. However, NPCs or other primary cells may not be the best system for the comparison. We think using an established cell line might be more informative, to control for effects such as 4-thiouridine toxicity.

**Recommendations for the authors:**

**Reviewer #1 (Recommendations for the authors):**
(1) The Western blot signals for SLC38A2 and ID2 are close to the membrane background and little convincing. Size markers are missing.

We agree these antibodies are not great. They are the best we could find, unfortunately. We have included originals of all western blots and gels as supplementary information. It’s important to note that the Riboseq data for ID2 and SLC38A2 are consistent with the western blots. See Figure 3C and Figure 3–figure supplement 3B.

(2) Figure 1 - Figure Supplement 1 appears to present data from a single experiment. This is far less than ideal considering the minor differences measured.

Thanks for the comment. This is a representative experiment showing the early time course. We have added a second experiment with two different treatments that show the same pattern in the puromycin assay, in Figure 1–figure supplement 1.

(3) Figure 3F: One wonders what this would look like if TE was plotted instead of RPF. Figure 3 - Figure Supplement 4 seems to show something along those lines. However, the data are not mentioned in the main results section are quite unclear. Why are data separated into TE high and low? Doesn't TE high in differentiated cells equal TE low in undifferentiated cells?

This is an interesting question. Note that in Figure 3B, n=6300 genes show no change in TE upon differentiation, compared to a total of n=2127 that show a change in TE, with most of those changes not very large. We have now replotted Figure 3F comparing irCLIP read counts in 3’-UTRs to RPF read counts, which shows a significant positive correlation, regardless of whether we look at undifferentiated or differentiated NPCs (See Figure 3F and a new Figure 3– figure supplement 4A). We also compare irCLIP reads in 3’-UTRs to TE values, which show no correlation (See Figure 3G and Figure 3–figure supplement 4B).

Figure 3-figure supplement 4 was actually a response to a previous round of review (at PLOS Biology) to a rather technical question from a reviewer. We think this figure and associated text should be removed. Instead, we now include supplementary tables with the processed RPF and TE values, for reference (Supplemental files 4-6). We omitted these in the original submission when they should have been included. We also abandoned comparing undifferentiated and differentiated NPCs, and instead look directly at irCLIP reads vs. RPFs or TE, regardless of NPC state, as noted above (Figure 3F, G, and Figure 3–figure supplement 4).

(4) Figure 3C: The data should be plotted on the same y-axis scale. This would make a visual assessment of the differences in mRNA and RFP levels more intuitive.

Thanks for this suggestion. We have rescaled the plots as requested.

**Reviewer #2 (Recommendations for the authors):**
(1) The quality of the Western blots in several figures is quite poor. Notably, Figure 1C seems to be a composite gel, as each blot appears to come from a different gel. Additionally, in Supplementary Figure 1A, there is only a single data point, yet the authors indicate that this image is representative of multiple assays. The lack of error bars in this figure raises a question vis-a-vis the reproducibility of the experiments.

Thanks for the comments. We now include all the original gels as supplementary information. As noted above, the antibodies for ID2 and SLC38A2 are not great, we agree. And as we noted above, the Riboseq data for ID2 and SLC38A2 are consistent with the western blots.

(2) For the top 500 targets of undifferentiated and differentiated NPCs in the Quick-irCLIP assay, the manuscript does not clarify how many targets are common and how many are unique to each condition. This information is important for understanding the extent of overlap and differentiation-specific interactions of eIF3 with mRNAs. Providing this data would strengthen the interpretation of the results.

There are 449 of the top 500 hits in common between undifferentiated and differentiated NPCs. We have now added this information to the text, to add clarity.

(3) The manuscript does not provide detailed percentages or numbers regarding the overlap between iCLIP and APA-Seq peaks. Clarifying this overlap, particularly in terms of how many of the APA sites are also targets of eIF3, would bolster the understanding of how these two datasets converge to support the authors' conclusions.

This is a difficult calculation to make, due to the fact that APA-Seq reads are generally much longer than the Quick-irCLIP reads. This is why we focused instead on quantifying the percent of Quick-irCLIP peaks (which are more narrow) overlap with predicted polyadenylation sequences, in Figure 2-figure supplement 1.

**Reviewer #3 (Recommendations for the authors):**
(1) Perform Quick-irCLIP in HEK293 cells to infer technical limitations and/or to generalize the model. The authors will then compare again eIF3 binding site in Jurkat, HEK293, and NPC.

This is an experiment we plan to do for a future publication, given that we would want to repeat both Quick-irCLIP and PAR-CLIP at the same time.

(2) Select mRNA candidates with high or low TE changes and analyze eIF3 binding and RPF density and protein abundance along NPC differentiation to support the role of eIF3 binding in stimulating translation.

We agree looking at time courses in more depth would be interesting. However, this would require substantial experimentation, which is better suited to a future study. Furthermore, now that we have moved away from comparing undifferentiated NPCs and differentiated NPCs when examining TE and RPF values (Figure 3 and Figure 3–figure supplement 4), we think the results now support a more general mechanism of translation reflected in the irCLIP 3’-UTR vs. RPF correlation, independent of NPC state.

(3) Analyze the interaction of eIF3 with eIF4G and other known partners. This will really provide an improvement to the manuscript. The lack of interaction between eIF3 and the 40S is quite surprising.

We agree more work needs to be done on the mechanistic side. These are experiments we think would be best to carry out in a stable cell line in the future, rather than primary cells.

(4) Perform Oligo-dT pulldown (or cap column if possible) and analyze the relative association of PABP, eIF3, and eIF4F on mRNA in NPC versus HEK293. This will clarify whether this mechanism of mRNA translation is specific to NPC or not.

Thanks for this suggestion. We are uncertain how it would be possible to deconvolute all the possible ways to interpret results from such an experiment. We agree thinking about ways to study the mechanism will keep us occupied for a while.

(5) Citations in the text indicate the first author, whereas the references are numbered!

Our apologies for this oversight. This was a carryover from previous formatting, and has been fixed.